# Unveiling the Spatial-temporal Effective Receptive Fields of Spiking Neural Networks

**Jieyuan Zhang**[1], **Xiaolong Zhou**[1], **Shuai Wang**[1] **Wenjie Wei**[1], **Hanwen Liu**[1],
**Qian Sun**[1], **Malu Zhang**[1,3]*, **Yang Yang**[1], **Haizhou Li**[2,3]

[1]University of Electronic Science and Technology of China,

[2]The Chinese University of Hong Kong, Shenzhen ,

[3]Shenzhen Loop Area Institute

## Abstract

Spiking Neural Networks (SNNs) demonstrate significant potential for energy-efficient neuromorphic computing through an event-driven paradigm. While training methods and computational models have greatly advanced, SNNs struggle to achieve competitive performance in visual long-sequence modeling tasks. In artificial neural networks, the effective receptive field (ERF) serves as a valuable tool for analyzing feature extraction capabilities in visual long-sequence modeling. Inspired by this, we introduce the Spatio-Temporal Effective Receptive Field (ST-ERF) to analyze the ERF distributions across various Transformer-based SNNs. Based on the proposed ST-ERF, we reveal that these models suffer from establishing a robust global ST-ERF, thereby limiting their visual feature modeling capabilities. To overcome this issue, we propose two novel channel-mixer architectures: multi-layer-perceptron-based mixer (MLPixer) and splash-and-reconstruct block (SRB). These architectures enhance global spatial ERF through all timesteps in early network stages of Transformer-based SNNs, improving performance on challenging visual long-sequence modeling tasks. Extensive experiments conducted on the Meta-SDT variants and across object detection and semantic segmentation tasks further validate the effectiveness of our proposed method. Beyond these specific applications, we believe the proposed ST-ERF framework can provide valuable insights for designing and optimizing SNN architectures across a broader range of tasks. The code is available at ○ EricZhang1412/Spatial-temporal-ERF.

## 1 Introduction

Spiking Neural Networks (SNNs) [1, 2] have emerged as a prominent research focus, characterized by binary spike activation that offers high sparsity, event-driven processing [3, 4], and biological plausibility [5]. Recent advances in encoding schemes [6, 7, 8], training methodologies [9, 10], and neuromorphic hardware [11, 12, 13] have enabled SNNs to achieve remarkable success in diverse tasks, including image processing [14, 15, 16], point/event analysis [17, 18], language understanding [19, 20, 21], and speech processing [22, 23, 24]. Nonetheless, SNNs still struggle to achieve performance comparable to their Artificial Neural Networks (ANNs) counterparts in visual long-sequence modeling tasks.

Compared to conventional image classification, visual long-sequence modeling tasks [25, 26] demand spatially dense outputs with prediction scales several orders of magnitude higher. This paradigm requires architectures capable of modeling long-range spatial dependencies, which are essential for

---

*Corresponding author: ✉maluzhang@uestc.edu.cn

39th Conference on Neural Information Processing Systems (NeurIPS 2025).

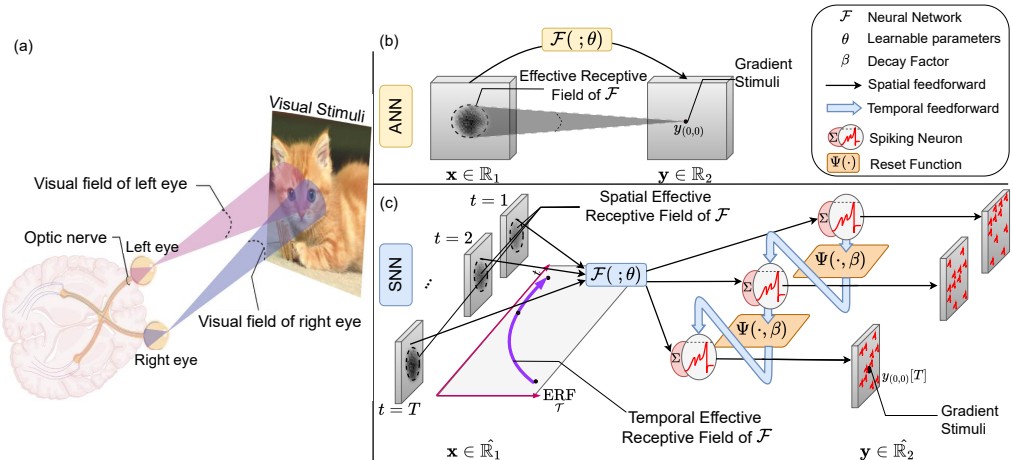

Figure 1: (a) The human visual field. (b) ERF in ANNs. (c) ST-ERF in SNNs. It extends the ERF to the temporal dimension, thus facilitating a comprehensive analysis of feature extraction in SNNs.

achieving competitive performance [25]. The Transformer [27] architecture introduces a self-attention mechanism that enables effective modeling of long-range spatial dependencies [28, 29, 30]. Motivated by this, recent studies have proposed various Transformer-based SNNs [31, 32, 33, 34], achieving notable progress in visual long-sequence tasks [15]. However, simply combining Transformers with SNNs may lead to suboptimal designs without fully considering the intrinsic spatio-temporal dynamics of spiking neurons. To bridge this gap, a more structured and interpretable framework is required to examine how SNNs model spatio-temporal dependencies. In this context, receptive field (RF) analysis offers a concrete lens through which their feature extraction capacity and attention allocation can be theoretically characterized.

In neuroscience, the RF represents the region of sensory input that can modulate a neuron's activity [35]. Borrowing this concept, the deep learning community defines a neuron's RF as the region of the input that can influence its output [36], with its size determined by network topology. However, this topology-based definition treats all positions within the RF equally, ignoring the learnable weights that determine the actual contribution of each input location. To refine this problem, researchers proposed the effective receptive field (ERF) [37] to quantify input features' contributions to output features via gradient analysis. Unlike topology-based RF, gradient-based ERF provides a more faithful characterization of the network's feature extraction patterns. However, such framework cannot be directly applied to SNNs due to the intrinsic spatio-temporal dynamics of spiking neurons. Therefore, we introduce the Spatial-Temporal Effective Receptive Field (ST-ERF) framework, which quantifies input feature contributions across spatio-temporal locations to characterize SNNs' feature extraction patterns. By jointly modeling temporal dependencies and spatial relationships, ST-ERF facilitates a comprehensive analysis of information processing in SNNs.

Based on the proposed ST-ERF, we analyze various Transformer-based SNNs and identify that existing models fail to establish effective global receptive fields across all timesteps. This limitation stems from the prevalent use of convolutional channel-mixers, which inherently introduce locality bias [38]. Despite facilitating efficient local feature extraction and long-range sparse modeling, this architectural design fundamentally constrains the long-range dense spatial interactions necessary for effective visual long-sequence modeling in SNNs. Building on these insights, we propose two novel channel-mixer architectures: multi-layer-perceptron-based mixer (MLPixer) and splash-and-reconstruct block (SRB). These designs use pixel-wise MLPs to keep spatial features separate when mixing channels, which reduces locality bias and improves the global receptive field in early stages of Transformer-based SNNs. Extensive experiments demonstrate the effectiveness of our methods on visual long-sequence tasks. Specially, on COCO 2017 object detection and ADE20K semantic segmentation, our Meta-SDT-Base [33] with SRB achieves 48.9% $\text{AP}_{50}^{\text{b}}$ and 43.7% mIoU, respectively, while maintaining a smaller model size. These results surpass state-of-the-art Transformer-based SNNs, thereby validating our ST-ERF analysis and further advancing SNNs in visual long-sequence modeling. Our main contributions are listed as follows:

- We propose the ST-ERF framework, extending the traditional ERF concept to the temporal dimension with rigorous mathematical formalization. ST-ERF systematically quantifies how input features at different spatial and temporal locations contribute to output features. This provides a theoretical tool for understanding and optimizing feature extraction in SNNs.

- We analyze various Transformer-based SNNs using the ST-ERF framework, revealing a critical limitation that existing models fail to establish a global ERF across all timesteps. To overcome this issue, we introduce two novel channel mixer designs: MLPixer and SRB, enabling Transformer-based SNNs to fully exploit their global modeling potential of long-range dependencies.

- We conduct extensive experiments on visual long-sequence modeling tasks and demonstrate that our method achieves superior performance. For instance, our Meta-SDT-Base with SRB achieves 48.9% $\text{AP}^{\text{b}}_{50}$ on COCO 2017 detection and 43.7% mIoU on ADE20K segmentation. It significantly outperforms existing state-of-the-art Transformer-based SNNs while using a smaller model size. These results strongly support the validity of our ST-ERF theoretical analysis and demonstrate the effectiveness of the proposed architectural designs.

## 2 Related Works

### 2.1 Receptive Field in Neural Networks

The human visual system perceives the external world through the visual fields of both eyes. As illustrated in Figure 1(a), each eye covers a specific region of the visual space, and these regions partially overlap in the center to enable binocular vision. Neurons along the visual pathway respond selectively to stimuli within their RFs. Over the past decades, the RF theory has profoundly influenced our understanding of how the brain filters and integrates visual information across spatial locations [39]. Inspired by the RF theory, deep neural networks adopt a similar principle by characterizing hierarchical ERFs that capture progressively abstract representations of input data. As shown in Figure 1(b), the ERF [37] formalizes this process by analyzing how spatial stimuli contribute to network activations. ERF has motivated extensive research at different architectural levels, from understanding basic operators [40] to designing higher-level modules and network structures such as Adaptive Receptive Fields [41], RF-Next [42], and AutoRF [43]. RF-based analysis has also driven advances in computational efficiency for lightweight architectures, influencing the development of CNN-based MobileNet variants [44, 45, 46] and MLP-based networks such as MLP-Mixer [47] and TSMixer [48]. Building on these insights, this work extends the ERF concept to SNNs, offering a theoretical framework for analyzing and optimizing their spatio-temporal feature extraction processes.

### 2.2 Visual Long-sequence Modeling in SNNs

Visual long-sequence modeling refers to tasks that require multiple predictions per image, rather than a single-label classification [49]. These tasks mainly include detection, segmentation, video understanding, and so on [25]. As these tasks involve modeling complex spatial and temporal dependencies, they demand architectures capable of capturing long-range contextual information. Transformer has become the dominant paradigm for visual long-sequence modeling owing to its global self-attention mechanism and flexible scalability. However, such models still suffer from high computational costs, primarily due to the quadratic complexity of self-attention, dense prediction requirements, and high-resolution inputs [50]. Recently, leveraging the sparse spike-driven nature of SNNs has emerged as a promising direction to mitigate these computational costs. Spike-driven Transformer series [51, 33, 52] adapt the standard Metaformer into an SNN framework for object detection and semantic segmentation, demonstrating the feasibility of SNNs in dense prediction tasks. Spike2Former [15] integrates normalized integer leaky-and-integrated firing (NI-LIF) neurons and spike-driven deformable attention to achieve competitive performance on segmentation benchmarks while maintaining low energy consumption. Despite these advancements, SNNs still lag behind ANNs in visual long-sequence modeling. This underscores the need for deeper investigation into SNNs' spatio-temporal bottlenecks and architectural optimization.

# 3 Theoretical Analysis of Spatio-temporal Effective Receptive Field

In this section, we first introduce the concept of ERF in conventional ANNs. Subsequently, we extend this conventional ERF into the temporal dimension to characterize the ST-ERF in SNNs. Finally, we introduce a loss-derived method to efficiently compute ST-ERF in SNNs.

## 3.1 ERF in ANNs

The concept of the ERF has been widely adopted to analyze how input features contribute to network activations and how such influences are distributed within the RF [40, 41]. Under the assumption of a single channel per layer, Luo et al. [37] mathematically characterized how each input feature contributes to the output of a neural network layer. It can be defined as follows:

$$\text{ERF}_{(i,j)}[y_{(m,n)}; \mathbf{x}] = \frac{\partial y_{(m,n)}}{\partial x_{(i,j)}}, \tag{1}$$

where $\mathbf{x} \in \mathbb{R}_1$ is the input feature and $\mathbf{y} \in \mathbb{R}_2$ is the output feature. In this manner, the ERF measures the partial derivative of an output feature $y_{(m,n)} \in \mathbf{y}$ with respect to each input feature $x_{(i,j)} \in \mathbf{x}$ within a given layer. As illustrated in Figure 1(b), the ERF of a given network $\mathcal{F}(; \theta)$ describes the input regions that contribute to a particular output activation.

As shown in Eq. (1), the ERF can be computed at any output location. However, most studies evaluate the ERF at the central output feature $y_{(0,0)}$ by assigning a unit gradient to this location [53, 54]. This practice establishes a centered and symmetric reference, ensuring stable and comparable visualization results. In this work, we also follow the setting of [37] and adopt the ERF at the central output feature $y_{(0,0)}$ as the evaluation metric.

## 3.2 ST-ERF in SNNs

Due to the inherent temporal dynamics, SNNs require additional consideration of the input at each timestep. To address this, we formally define the ST-ERF (i.e., $\text{ERF}^{(\mathcal{S},\mathcal{T})}$). Firstly, we redefine the mapping relationship of SNNs. Consider a SNN layer with learnable parameters $\theta$ that maps input spike features $\mathbf{x}[1:T] \in \hat{\mathbb{R}}_1$ to output spike features $\mathbf{y}[1:T] \in \hat{\mathbb{R}}_2$:

$$\mathbf{y}[1:T] = \mathcal{F}(\mathbf{x}[1:T]; \theta), \mathcal{F} : \hat{\mathbb{R}}_1 \to \hat{\mathbb{R}}_2. \tag{2}$$

Its ERF needs to account not only for the accumulation across spatial dimensions but also for that across temporal dimensions. Specifically, $\text{ERF}^{(\mathcal{S},\mathcal{T})} \in \hat{\mathbb{R}}_1$ can be expressed as:

$$\text{ERF}^{(\mathcal{S},\mathcal{T})}_{(i,j)}[y_{(m,n)}[t], \tau; \mathbf{x}] = \frac{\partial y_{(m,n)}[t]}{\partial x_{(i,j)}[t-\tau]}, 1 \le t \le T, 0 \le \tau \le t-1. \tag{3}$$

Accordingly, $\text{ERF}^{(\mathcal{S},\mathcal{T})}$ quantifies how much each input feature $x_{(i,j)}[t-\tau] \in \mathbf{x}$ at a previous timestep $t - \tau$ contributes to a specific output feature $y_{(m,n)}[t] \in \mathbf{y}$. Based on this definition, the spatial ERF (i.e., $\text{ERF}^{(\mathcal{S})}$) can be seen as the weighted average of the ST-ERFs over all timesteps:

$$\text{ERF}^{(\mathcal{S})}_{(i,j)}[y_{(m,n)}; \mathbf{x}] = \frac{1}{T} \sum_{t=1}^{T} \sum_{\tau=0}^{t-1} w(t, \tau) \cdot \text{ERF}^{(\mathcal{S},\mathcal{T})}_{(i,j)}[y_{(m,n)}[t]; \mathbf{x}, \tau], \tag{4}$$

where $w(t, \tau)$ represents the relative contribution of the input with delay $\tau$ at time $t$ to the output. The specific form of $w(t, \tau)$ depends on the neuronal dynamics and network architecture. For example, in Leaky Integrate-and-Fire (LIF) neurons, inputs closer to the current time step may have a higher influence due to the decay of membrane potential over time.

The temporal ERF (i.e., $\text{ERF}^{(\mathcal{T})}$) can be seen as the integration over the spatial dimensions of ST-ERF to indicate the contribution of inputs at different timesteps to the final output:

$$\text{ERF}^{(\mathcal{T})}[\tau; \mathbf{x}] = \sum_{i,j} \sum_{m,n} \text{ERF}^{(\mathcal{S},\mathcal{T})}_{(i,j)}[y_{(m,n)}[T]; \mathbf{x}, \tau]. \tag{5}$$

As shown in Figure 1(c), we visualize an example of the ST-ERF. Similar to conventional ERF analysis, we focus on the center of the feature map at a specific timestep (e.g., the final timestep in Fig. 1(c)) to analyze the spatio-temporal feature representations in an SNN. Depending on the purpose of analysis, one may investigate the spatial distribution of the ST-ERF at a given timestep (spatial ERF) or its temporal distribution across one or more layers (temporal ERF).

### 3.3 Loss-Derived Calculation for ST-ERFs

Based on Eq. (3), computing the ST-ERF in SNNs requires evaluating first-order derivatives of outputs with respect to all input features. To obtain the ST-ERF conveniently, we introduce the loss-derived calculation method to efficiently compute using PyTorch's Automatic Differentiation functionality. Consider a SNN with input spike features $s^{\ell-1}$, output spike features $s^\ell$ at the $\ell$-th layer, and an arbitrary loss function $\mathcal{L}$. The spatial ERF of SNNs can be easily obtained by calculating the average of the gradient of the loss with respect to input features at position $(i, j)$ across all timesteps $T$. Specifically, it can be computed as follows:

$$\text{ERF}^{(\mathcal{S})}_{(i,j)}[s^\ell_{(0,0)}] = \frac{1}{T}\sum_{t=1}^{T}\frac{\partial s^\ell_{(0,0)}[t]}{\partial s^{\ell-1}_{(i,j)}[t]} = \frac{1}{T}\sum_{t=1}^{T}\frac{\partial \mathcal{L}}{\partial s^{\ell-1}_{(i,j)}[t]}, \text{when} \forall t, \frac{\partial \mathcal{L}}{\partial s^\ell_{(\hat{i},\hat{j})}[t]} = \begin{cases} 1, & \hat{i}=0, \hat{j}=0, \\ 0, & \text{otherwise} \end{cases}.$$
(6)

The temporal ERF of SNNs can be obtained by calculating the sum of the gradient of the loss function with respect to input features at timestep $T - \tau$ across all spatial positions. It can be computed as:

$$\text{ERF}^{(\mathcal{T})}[\tau] = \sum_{i,j}\sum_{\hat{i},\hat{j}}\frac{\partial s^\ell_{(\hat{i},\hat{j})}[T]}{\partial s^{\ell-1}_{(i,j)}[T-\tau]} = \sum_{i,j}\frac{\partial \mathcal{L}}{\partial s^{\ell-1}_{(i,j)}[T-\tau]}, \quad \text{when } \forall \hat{i}, \hat{j}, \frac{\partial \mathcal{L}}{\partial \mathbf{s}^\ell_{(\hat{i},\hat{j})}[T]} = 1. \quad (7)$$

Proof can be found in Appendix A. We refer to the conditions in Eq. (6) and (7) as gradient stimuli. Based on this proposition, we could easily obtain the spatial and temporal ERF with automatic back-propagation without an explicit loss function.

## 4 Problem Analysis on Transformer-based SNNs using ST-ERF

In this section, we use the ST-ERF framework to analyze existing Transformer-based SNNs and identify their limitations in visual long-sequence modeling tasks.

### 4.1 Different ST-ERF Behaviors in Transformer-based SNNs

We apply the ST-ERF framework to analyze Transformer-based SNNs' spatial ERF behaviors across all timesteps. Specifically, we compared two groups of architectures(a: ViT-like architecture group and b: Meta-architecture group) with their ANN counterparts to investigate the differences in the formation of their spatial ERFs. For the loss-derived calculation, we set the central patch across all channels and timesteps in the output tensor as the gradient stimuli (uniform values of 1), then perform automatic back-propagation. Each experiment comprised 60 iterations using randomly sampled input tensors under standard normal distribution ($\mu = 0, \sigma^2 = 1$). Note that we average the ST-ERF over all timesteps to obtain a clear visualization.

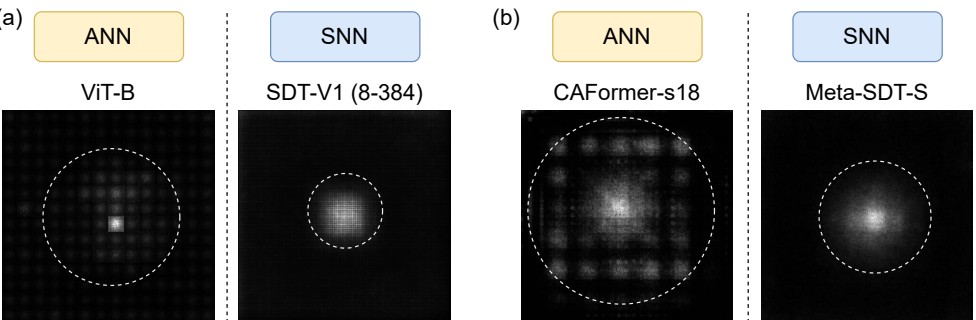

Figure 2: Comparison of spatial ERF with ANN Vision Transformers and ST-ERF with different Transformer-based SNNs. (a) ViT-like architecture comparison group: ViT-B and SDT-V1. (b) Meta-architecture comparison group: CAFormer-s18 and its counterpart Meta-SDT-S.

The comparison of ViT-like architectures is illustrated in Figure 2(a). Compared with the classic ViT-B, SDT-V1 exhibits a more centrally concentrated yet markedly narrower spatial ERF. This

observation suggests that the architectural modifications in SDT-V1 may restrict the receptive field's spatial extent, thereby enhancing its attention on local spatial dependencies. In fact, SDT-V1 adopts a fundamentally different strategy from the vanilla ViT in the patch splitting stage. Specifically, SDT-V1 employs the Spike Patch Splitting (SPS) module, consisting a Patch Splitting Module (PSM) to linearly project the input image and a Relative Position Embedding (RPE) [55] block to generate the latent position information. The SPS module incorporates multiple convolutional layers at the early stage of the network, facilitating low-level spatial features extraction from input images.

The comparison of meta-architectures is illustrated in Figure 2(b). Although Meta-SDT exhibits ERF behaviors similar to those of its ANN counterparts, it also struggles to maintain long-range feature attention. This limitation can be attributed to the additional employment of convolutional layers, which tend to emphasize localized features rather than global spatial contexts. Compared with CAFormer, Meta-SDT introduces the Re-parameterization Convolution (RepConv)[56] to perform the linear projection of queries, keys, and values[51]. This design enhances local feature extraction, yet it inherently constrains the model's capacity to aggregate information across distant spatial regions. Together, these findings suggest that the convolutional operations enhances local feature sensitivity but poses challenges for maintaining long-range spatial coherence in Transformer-based SNNs.

## 4.2 Visual Long-sequence Modeling Needs Global ST-ERF

Visual long-sequence modeling tasks often involve dense predictions across an entire image, requiring the processing of thousands of input tokens [25]. Therefore, capturing long-range dependencies and global context is crucial for achieving accurate and robust representations [57]. Prior studies have found that vision models with global receptive fields often excel at segmentation and detection, for instance when using self-attention mechanisms as in Transformer architectures [57, 58]. In contrast, architectures lacking global context integration tend to struggle. While early convolutional layers excel at extracting low-level structural patterns [59], their locality inherently limits the capacity to capture long-range dependencies, making them suboptimal for visual long-sequence tasks.

However, despite the need for global spatial awareness in visual long-sequence modeling, Transformers-based SNNs still fail to achieve a truly global ST-ERF. As discussed above, they tend to focus heavily on the center and expand to limited size. This contrasts sharply with the expected behavior required for visual long-sequence modeling tasks, where the weak global ST-ERF limits information aggregation and consequently degrades performance on such scenarios [57].

# 5 Methods

In this section, we propose two novel channel-mixing designs, MLPixer and SRB, which enable Transformer-based SNNs to more effectively capture long-range dependencies. Furthermore, we integrate these modules into the Meta-SDT architecture to enhance performance on visual long-sequence modeling tasks.

## 5.1 Design of Channel Mixer Block

To enhance the global modeling capability of SNN in visual long-sequence tasks, we propose two novel channel mixer designs. The first is the multi-layer perceptron-based mixer (MLPixer), which employs a two-layer MLP structure to more effectively extract global features. It is defined as follows:

$$\text{MLPixer}(\mathbf{X}) = \text{BN}\Big(\text{MLP}\big(\mathbb{SN}\{\text{BN}(\text{MLP}\{\mathbb{SN}(\mathbf{X})\})\}\big)\Big), \tag{8}$$

where $\mathbf{X} \in \mathbb{R}^{T \times B \times N \times D}$ denotes the input of channel mixers in the Transformer block. $\mathbb{SN}(\cdot)$ denotes a spiking neuron layer that transforms the input sequence into the spike trains. $\text{MLP}(\cdot)$ denotes a single-layer fully connected (FC) operation, and $\text{BN}(\cdot)$ denotes batch normalization.

Compared with vanilla channel mixers [60, 51] that rely on convolution operations, the MLPixer employs a two-layer MLP operation to mix features across channels. This design reduces reliance on convolutional operations, mitigating the ERF's bias toward a Gaussian-like central concentration and enabling SNNs to capture long-range dependencies more effectively.

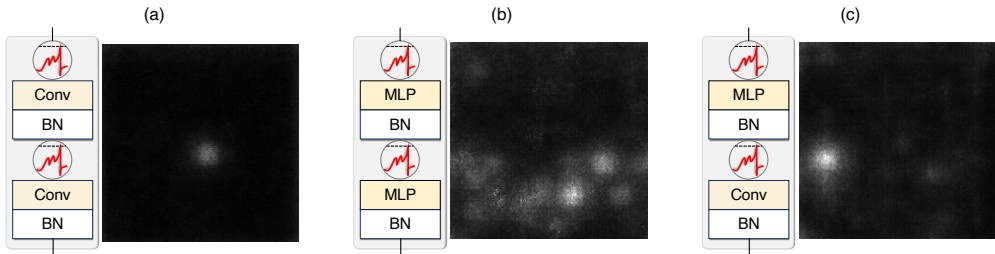

Figure 3: Comparison between the original channel mixer design and our proposed methods, along with their ST-ERF. For clearer visualization, the ST-ERF is averaged over timesteps. (a) Vanilla convolution-based channel mixer. (b) Proposed MLPixer architecture. (c) Proposed SRB architecture. Obviously, the vanilla convolution-based channel mixer exhibits a limited ST-ERF, whereas our MLPixer and SRB modules achieve a more global ST-ERF. Moreover, due to the reduced use of convolutions, MLPixer exhibits an even broader effective receptive field.

Building on this, we further propose the SRB module. It replaces only the second convolution in the channel mixer with a single-layer MLP operation. Specifically, the SRB is defined as follows:

$$\text{SRB}(\mathbf{X}) = \text{BN}\Big(\text{MLP}\big(\mathbb{SN}\{\text{BN}(\text{Conv}\{\mathbb{SN}(\mathbf{X})\})\}\big)\Big). \tag{9}$$

Here, $\text{Conv}(\cdot)$ denotes a $1\times1$ convolution operation. In this manner, SRB module reduces additional parameters while maintaining performance. To validate the effectiveness of our approach, we visualize the ERFs of the Conv-based mixer, the MLPixer, and the SRB modules.

As shown in Figure 3(a), the vanilla convolution-based channel mixer exhibits a limited ST-ERF. In contrast, the proposed MLPixer and SRB modules demonstrate a more global ST-ERF. Furthermore, the comparison between Figure 3(b) and Figure 3(c) further demonstrates that MLPixer exhibits a more global ERF. This stems from reduced use of convolutions and further suggests that MLPs provide stronger global modeling capacity than convolutional operators. We will validate the proposed module on visual long-sequence modeling tasks in the experiment section.

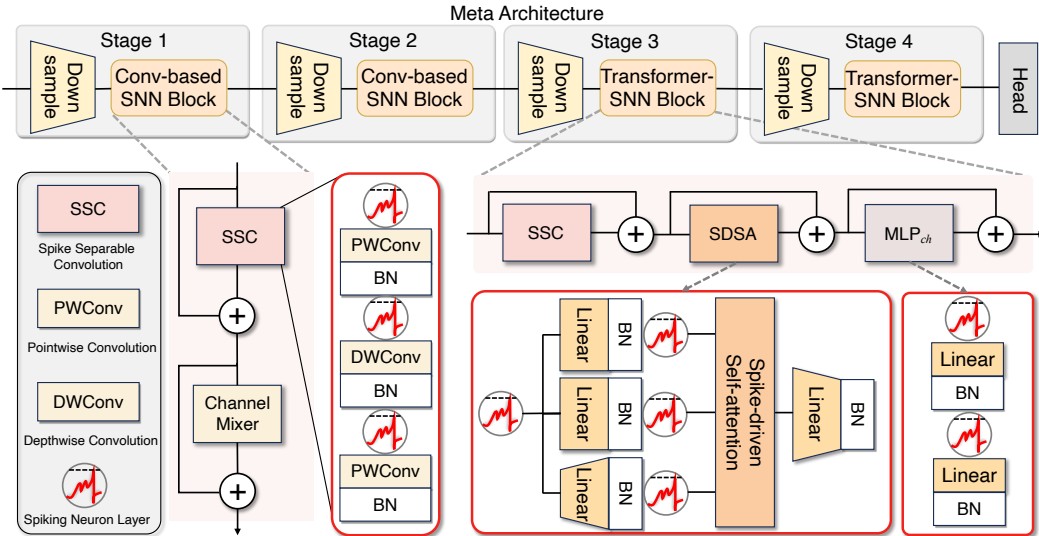

Figure 4: The overall architecture of Meta-SDT, which typically comprises four hierarchical stages. The first two stages use convolution-based SNN blocks, while the latter two adopt Transformer-SNN blocks. To strengthen the global modeling capacity of SNNs, we introduce two novel channel mixer architectures, MLPixer and SRB, to replace the convolution-based SNN blocks in the first two stages.

## 5.2 Overall Architecture

To further validate the effectiveness of our approach, we integrate the proposed SRB and MLPixer modules into the CAFormer [60] and Meta-SDT [33] architectures. As shown in Figure 4, these architectures adopt a multi-stage design, where the first two stages consist of Conv-based SNN blocks, and the latter two stages comprise Transformer-SNN blocks. In this work, we replace only the operations in the first two stages. Specifically, the first two stages are represented as:

$$\mathbf{X}' = \mathbf{X} + \text{SSC}(\mathbf{X}), \mathbf{X}'' = \mathbf{X}' + \text{Mixer}_\epsilon(\mathbf{X}'). \tag{10}$$

Here, $\text{Mixer}_\epsilon(\cdot)$ is the channel mixer. In this work, we implement the approach using both the MLPixer module and the SRB module. Similar to the vanilla channel mixer, our method adopts an up-projection followed by a down-projection with a nonlinear activation in between, where $\epsilon > 1$ represents the intermediate dimensional expansion ratio. $\text{SSC}(\cdot)$ is the spike-driven separable convolution block as token mixer, it is defined as follows:

$$\text{SSC}(\mathbf{X}) = \text{PWConv}_2(\mathbb{SN}(\text{DWConv}(\mathbb{SN}(\text{PWConv}_1(\mathbb{SN}(\mathbf{X})))))) . \tag{11}$$

$\text{PWConv}_1(\cdot)$ and $\text{PWConv}_2(\cdot)$ are pointwise convolutions, $\text{DWConv}(\cdot)$ is depthwise convolution. $\mathbb{SN}(\cdot)$ denotes the spiking neuron layer. To maintain the spike-driven characteristics of the network, we implement membrane-shortcut residual connection mechanism. Furthermore, Transformer-SNN blocks are utilized in Stage 3 and Stage 4, following the same configuration as that of Meta-SDT-V3 [33]. We will further verify the effectiveness of the proposed method in the experimental section.

# 6 Experiments

In this section, we validate the effectiveness of our method through visualization and experimental analysis. First, we examine the changes in the ST-ERF after integrating the proposed modules into Meta-SDT, showing that our method achieves stronger global spatial receptive fields across all stages. Second, we evaluate its performance improvement on long-sequence modeling tasks, including object detection and semantic segmentation. Finally, we further investigate the method on complex event modeling tasks to assess its applicability in more challenging scenarios.

## 6.1 ST-ERF Behavior in Transformer-based SNNs

In order to study the impact of our proposed block on the receptive field of Meta-SDT, we compared temporal-averaged spatial ERFs between our two Meta-SDT variants with previous models. We initialized the central spatial feature across all channels and timesteps in the output tensor as uniform gradient stimuli (value = 1), and propagated the gradients backward through the network. Each experiment consisted of 60 iterations with input tensors randomly drawn from a standard normal distribution ($\mu = 0, \sigma^2 = 1$).

The results are illustrated in Figure 5. Surprisingly, we found that Spikformer exhibits diffuse receptive fields across all stages. The SDT-V1, Meta-SDT, and QKFormer demonstrate markedly centered distribution that gradually expand as the network deepens, all manifesting a Gaussian-like effect. Additionally, we observed dissipation of spatial ERF in SDT-V1 during the final stage. In contrast, our proposed two Meta-SDT variants establish robust global spatial receptive fields in the early stages. The MLPixer-SDT establishes a strong global spatial ERF in Stage 1. As the network deepens, its spatial ERF selectively contracts toward specific regions. The ERF behavior in SRB-SDT is slightly different, as it only begins to form a preliminary spatial ERF at Stage 2, and this distribution continues to evolve with increasing network depth.

## 6.2 Performance in Visual Long-sequence Modeling Tasks

We selected two challenging datasets to evaluate performance on classic visual long-sequence modeling tasks: object detection and instance segmentation on COCO 2017, and semantic segmentation on the ADE20K dataset. We choose the Meta-SDT(v3) [33] as the baseline and construct Meta-SDT variants with MLPixer($\epsilon$4), MLPixer($\epsilon$6) and SRB($\epsilon$4).

**Performance on COCO 2017** We evaluate the efficacy of the MLPixer and SRB on Meta-SDT and select the classic and large-scale COCO [61] dataset as our benchmark for evaluation. Following

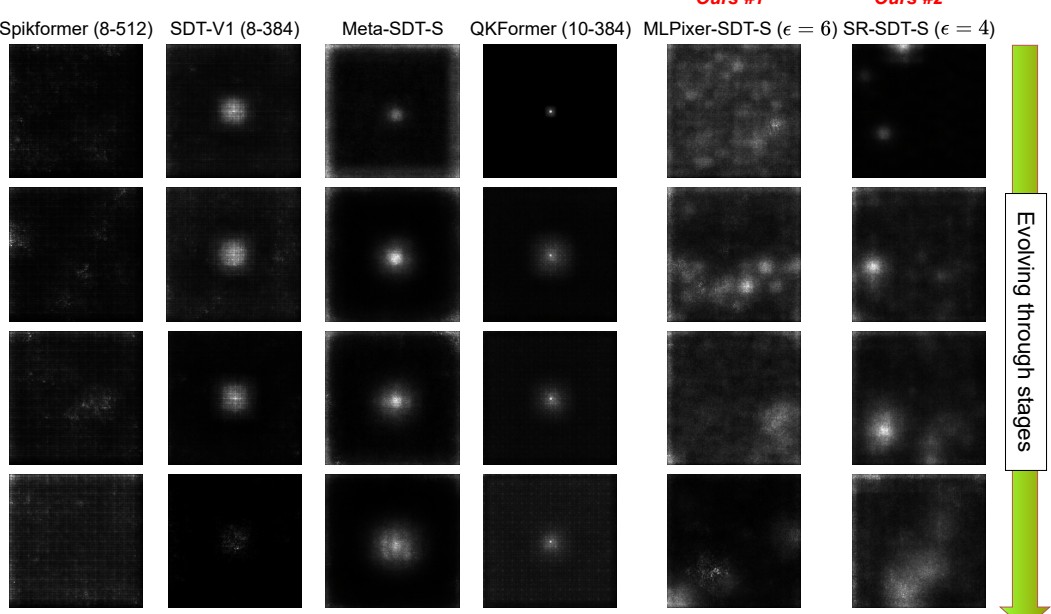

Figure 5: Comparison of temporal-averaged spatial ERF evolution across stages. From top to bottom are Stage 1 through Stage 4. Spikformer shows diffuse receptive fields across all stages. SDT-V1, Meta-SDT, and QKFormer exhibit more centered spatial distributions that gradually expand as depth increases. Our two Meta-SDT variants establish global spatial receptive fields in the early stages.

the previous work [51, 33, 52], we use the MMdetection [62] codebase with a spiking version and then deploy our model. We employ the Meta-SDT [33] with two variants as the backbone network to extract features, along with fine-tuning Mask R-CNN [63] for object detection and instance segmentation. All backbone networks are pretrained on ImageNet-1K [64], while the incremental layers are initialized following [65]. During fine-tuning, we strictly obey the $1\times$ training schedule.

The comparison results of object detection and instance segmentation are shown in Table 1. Under the same training schedule, both the MLPixer and SRB variants outperforms the baseline across all metrics. More specifically, the SRB variant exceeds the performance of SDTv3-T and SDTv3-B by $10.42\%$ and $4.26\%$ on the $AP_{50}^b$ metric, while maintaining almost the same model size. In conclusion, our approach demonstrates efficacy in object detection and instance segmentation, setting a new benchmark for COCO dataset in the SNN domain.

**Performance on ADE20K** We evaluate the performance of MLPixer and SRB on the semantic segmentation task using the challenging ADE20K dataset [66]. Similar to the COCO experiments, we utilize the spiking version of MMSegmentation [67] as our codebase and employ the Meta-SDT [33] with two variants as the backbone network. We fine-tune the Semantic FPN framework [68] for semantic segmentation. Backbone networks are initialized with ImageNet-1K pre-trained weights [64], and new layers follow the initialization scheme of [65]. All models are strictly obey the same training schedule for 160k iterations.

As shown in Table 2, both MLPixer and SRB variants surpass the baseline in terms of mIoU. The SRB variant improves performance by 3.3% and 2.6% over SDTv3-T and SDTv3-B, respectively, while reducing parameters by 0.3M and 1.2M. The MLPixer($\epsilon$4) variant achieves the largest parameter reduction of 0.6M and 2.4M, with comparable or superior accuracy to SDTv3-T and SDTv3-B. These results highlight the effectiveness of the proposed modules in enhancing semantic segmentation on ADE20K.

### 6.3 Performance in Complex Event Modeling Tasks

**Event-based Tracking** We evaluate the effectiveness of two channel mixers in the context of event-based tracking, a highly challenging yet practically significant application domain for SNNs. Our experiments follow the SDTrack pipeline [18], which employs the Global Trajectory Prompt

| Arch. | #T | #P | $AP^b$ | $AP^b_{50}$ | $AP^b_{75}$ | $AP^m$ | $AP^m_{50}$ | $AP^m_{75}$ |
|---|---|---|---|---|---|---|---|---|
| SDTv3-T[33] | 4 | 25M | 15.2 | 35.5 | 10.2 | 15.2 | 33.0 | 12.3 |
| MLPixer($\epsilon$4) | 4 | 24M | 16.2 | 37.0 | 11.5 | 15.2 | 32.9 | 12.5 |
| MLPixer($\epsilon$6) | 4 | 25M | 17.5 | 38.5 | 13.2 | 16.2 | 34.5 | 13.5 |
| SRB($\epsilon$4) | 4 | 25M | **18.2** | **39.2** | **13.8** | **17.5** | **34.8** | **14.3** |
| SDTv3-B[33] | 4 | 39M | 21.7 | 46.9 | 17.0 | 20.1 | 41.8 | 17.5 |
| MLPixer($\epsilon$4) | 4 | 36M | 22.9 | 47.6 | 19.2 | 21.0 | 43.4 | 18.3 |
| MLPixer($\epsilon$6) | 4 | 39M | 25.1 | 48.8 | 22.5 | 21.9 | 43.5 | 19.6 |
| SRB($\epsilon$4) | 4 | 37M | **25.8** | **48.9** | **22.8** | **22.5** | **43.9** | **20.4** |

Table 1: Object detection and instance segmentation with Mask R-CNN on COCO val2017, using ImageNet-1K pre-training and $1\times$ training schedule.

| Arch. | Ch. Mixer | #T | Param.(M) | mIoU(%) |
|---|---|---|---|---|
| | C2d-k3($\epsilon$4) | 4 | 6.5 **BASE** | 34.9 **BASE** |
| SDTv3 | MLPix.($\epsilon$4) | 4 | 5.9 (↓0.6) | 34.9 (↑0.0) |
| -T[33] | MLPix.($\epsilon$6) | 4 | 6.6 (↑0.1) | 35.9 (↑1.0) |
| | SRB($\epsilon$4) | 4 | 6.2 (↓0.3) | **38.2** (↑3.3) |
| | C2d-k3($\epsilon$4) | 4 | 20.4 **BASE** | 41.1 **BASE** |
| SDTv3 | MLPix.($\epsilon$4) | 4 | 18.0 (↓2.4) | 42.0 (↑0.9) |
| -B[33] | MLPix.($\epsilon$6) | 4 | 20.7 (↑0.3) | 43.4 (↑2.3) |
| | SRB($\epsilon$4) | 4 | 19.2 (↓1.2) | **43.7** (↑2.6) |

Table 2: Segmentation results on ADE20K based on different mixer block, using ImageNet-1K pre-training and 160k iter.

method to convert event streams into event frames. We strictly adhere to the original training protocol, modifying only the backbone by replacing SDTrack with our proposed SDTrack+MLPixer or SDTrack+SRB variants. As presented in Table 3, extensive experiments on the FE108 [69] and VisEvent [70] datasets demonstrate that our architectures surpass the original SDTrack in several key metrics. These results confirm that both the MLPixer and SRB designs preserve the Transformers-based SNNs' performance, yet highlight opportunities for further improvement in subsequent temporal benchmarks.

Table 3: Performance comparison on event-based object tracking, a challenging yet important application for SNNs. Evaluation is conducted on two benchmark datasets, FE108 and VisEvent.

| Architecture | Timesteps | Param. (M) | FE108 [69] | | VisEvent [70] | |
|---|---|---|---|---|---|---|
| | | | AUC(%) | PR(%) | AUC(%) | PR(%) |
| SD-Track(Tiny) [18] | $4 \times 1$ | 19.61 | 56.7 | 89.1 | **35.4** | 48.7 |
| +MLPixer ($\epsilon = 4$) | $4 \times 1$ | 20.21 | 57.1 | 89.2 | 33.7 | 47.3 |
| +MLPixer ($\epsilon = 6$) | $4 \times 1$ | 22.99 | 57.9 | **90.1** | 34.5 | **48.9** |
| +SRB ($\epsilon = 4$) | $4 \times 1$ | 21.43 | **58.2** | 88.5 | 33.8 | 48.0 |

## 7 Conclusion

This paper presents ST-ERF as a novel framework for analyzing the spatial-temporal modeling behaviors in SNNs from a new perspective. Through this analysis, an inherent limitation in current Transformer-based SNN models is identified when applied to visual long-sequence modeling tasks. To address this limitation, two channel-mixer architectures, MLPixer and SRB, are proposed. Visualization of ST-ERF demonstrates that both modules enhance the global receptive field. Extensive experiments on long-sequence modeling tasks, including object detection and semantic segmentation, show that MLPixer and SRB improve overall performance, with SRB achieving an optimal balance between accuracy and model size. Furthermore, the study investigates complex event modeling tasks to assess the applicability of MLPixer and SRB in more challenging scenarios. Overall, the proposed ST-ERF framework offers valuable insights for the design and optimization of SNN architectures across a wide range of tasks.

## Acknowledgments

This work is supported in part by the National Natural Science Foundation of China (No. 62220106008 and 62271432), in part by the Shenzhen Science and Technology Program (Shenzhen Key Laboratory, Grant No. ZDSYS20230626091302006) and in part by the Program for Guangdong Introducing Innovative, Entrepreneurial Teams, Grant No. 2023ZT10X044, and in part by the State Key Laboratory of Brain Cognition and Brain-inspired Intelligence Technology, Grant No. SKLBI-K2025010. This work was partially supported by UESTC Kunpeng&Ascend Center of Cultivation.

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

# Appendix

## A  Proof of 3.3

*Proof.* Leaky Integrate-and-Fire (LIF) model [5, 2] can be described by the following equations:

$$\mathbf{v}^\ell[t] = \mathbf{h}^\ell[t-1] + f(\mathbf{w}^\ell, \mathbf{x}^{\ell-1}[t-1]), \qquad \text{(Charging function)}, \qquad (12)$$

$$\mathbf{s}^\ell[t] = \boldsymbol{\Theta}(\mathbf{v}^\ell[t] - \vartheta), \qquad \text{(Firing function)}, \qquad (13)$$

$$\mathbf{h}^\ell[t] = \begin{cases} \beta \mathbf{v}^\ell[t] - \vartheta \mathbf{s}^\ell[t], & \text{soft reset} \\ \mathbf{v}^\ell[t](1 - \mathbf{s}^\ell[t]), & \text{hard reset} \end{cases} \qquad \text{(Leak-and-reset function)}, \qquad (14)$$

where $\beta$ is the decay constant, $t$ is the time step, $\mathbf{w}^\ell$ is the weight matrix of layer $\ell$, $f(\cdot)$ is the operation that stands for convolution (Conv) or fully connected (FC), $\mathbf{x}$ is the input, and $\boldsymbol{\Theta}(\cdot)$ denotes the Heaviside step function. When the membrane potential $\mathbf{v}$ exceeds the firing threshold $\vartheta$, the LIF neuron will trigger a spike $\mathbf{s}$; otherwise, it remains inactive. After spike emission, the neuron invokes the reset mechanism, where the soft reset function is employed. $\mathbf{h}$ is the membrane potential following the reset function.

For the back-propagation of this neuron, we introduce the training process of SNN gradient descent and the parameter update method of spatio-temporal back-propagation (STBP) [9, 71]. The accumulated gradients of loss $\mathcal{L}$ with respect to weights $\mathbf{w}$ at layer $\ell$ can be calculated as:

$$\frac{\partial \mathcal{L}}{\partial \mathbf{w}^\ell} = \sum_{t=1}^{T} \frac{\partial \mathcal{L}}{\partial \mathbf{s}^{\ell+1}[t]} \frac{\partial \mathbf{s}^{\ell+1}[t]}{\partial \mathbf{v}^{\ell+1}[t]} \Big( \frac{\partial \mathbf{v}^{\ell+1}[t]}{\partial \mathbf{w}^\ell} + \sum_{\tau < t} \prod_{i=t-1}^{\tau} \Big( \frac{\partial \mathbf{v}^{\ell+1}[i+1]}{\partial \mathbf{v}^{\ell+1}[i]} + \frac{\partial \mathbf{v}^{\ell+1}[i+1]}{\partial \mathbf{s}^{\ell+1}[i]} \frac{\partial \mathbf{s}^{\ell+1}[i]}{\partial \mathbf{v}^{\ell+1}[i]} \Big) \frac{\partial \mathbf{v}^{\ell+1}[\tau]}{\partial \mathbf{w}^\ell} \Big), \tag{15}$$

where $\mathbf{s}^\ell[t]$ and $\mathbf{v}^\ell[t]$ represent the output spikes and membrane potential of the neuron in layer $\ell$, at time $t$. Moreover, notice that $\frac{\partial \mathbf{s}^\ell[t]}{\partial \mathbf{v}^\ell[t]}$ is non-differentiable. To overcome this problem, Wu et al. [9] propose the surrogate function to make only the neurons whose membrane potentials close to the firing threshold receive nonzero gradients during back-propagation.

In this paper, we use the rectangle function, which has been shown to be effective in gradient descent and may be calculated by:

$$\frac{\partial \mathbf{s}^\ell[t]}{\partial \mathbf{v}^\ell[t]} = \frac{1}{a} \operatorname{sign}\Big( \big| \mathbf{v}^\ell[t] - \vartheta \big| < \frac{a}{2} \Big), \tag{16}$$

where $a$ is a defined coefficient for controlling the width of the gradient window.

To compute $\sum_{t=1}^{T} \frac{\partial \mathbf{s}^\ell_{(0,0)}[t]}{\partial \mathbf{s}^{\ell-1}_{(i,j)}[t]}$, we follow the chain rule with an arbitrary loss $\mathcal{L}$. Consider $\sum_{t=1}^{T} \frac{\partial \mathcal{L}}{\partial \mathbf{s}^{\ell-1}_{(i,j)}[t]}$:

$$
\begin{aligned}
\sum_{t=1}^{T} \frac{\partial \mathcal{L}}{\partial \mathbf{s}^{\ell-1}_{(i,j)}[t]} &= \sum_{t=1}^{T} \sum_{\hat{i},\hat{j}} \frac{\partial \mathcal{L}}{\partial \mathbf{s}^\ell_{(\hat{i},\hat{j})}[t]} \frac{\partial \mathbf{s}^\ell_{(\hat{i},\hat{j})}[t]}{\partial \mathbf{s}^{\ell-1}_{(i,j)}[t]} \\
&= \sum_{\hat{i}} \sum_{\hat{j}} \sum_{t=1}^{T} \frac{\partial \mathcal{L}}{\partial \mathbf{s}^\ell_{(\hat{i},\hat{j})}[t]} \frac{\partial \mathbf{s}^\ell_{(\hat{i},\hat{j})}[t]}{\partial \mathbf{s}^{\ell-1}_{(i,j)}[t]} \\
&= \sum_{\hat{i} \neq 0} \sum_{\hat{j} \neq 0} \sum_{t=1}^{T} \frac{\partial \mathcal{L}}{\partial \mathbf{s}^\ell_{(\hat{i},\hat{j})}[t]} \frac{\partial \mathbf{s}^\ell_{(\hat{i},\hat{j})}[t]}{\partial \mathbf{s}^{\ell-1}_{(i,j)}[t]} + \sum_{t=1}^{T} \frac{\partial \mathcal{L}}{\partial \mathbf{s}^\ell_{(0,0)}[t]} \frac{\partial \mathbf{s}^\ell_{(0,0)}[t]}{\partial \mathbf{s}^{\ell-1}_{(i,j)}[t]}.
\end{aligned}
\tag{17}
$$

When the following conditions are met:

$$\forall t \in T, \frac{\partial \mathcal{L}}{\partial \mathbf{s}^\ell_{(\hat{i},\hat{j})}[t]} = \begin{cases} 1 & \hat{i} = 0, \hat{j} = 0, \\ 0 & otherwise \end{cases}. \tag{18}$$

We can get:

$$\sum_{t=1}^{T} \frac{\partial \mathcal{L}}{\partial \mathbf{s}_{(i,j)}^{\ell-1}[t]} = \sum_{t=1}^{T} \frac{\partial \mathbf{s}_{(0,0)}^{\ell}[t]}{\partial \mathbf{s}_{(i,j)}^{\ell-1}[t]}, \quad \frac{1}{T}\sum_{t=1}^{T} \frac{\partial \mathcal{L}}{\partial \mathbf{s}_{(i,j)}^{\ell-1}[t]} = \frac{1}{T}\sum_{t=1}^{T} \frac{\partial \mathbf{s}_{(0,0)}^{\ell}[t]}{\partial \mathbf{s}_{(i,j)}^{\ell-1}[t]}. \tag{19}$$

The spatial ERF at position $(i, j)$ can thus be calculated by summing the gradients of the loss with respect to all timesteps.

For the temporal ERF, we need to compute $\sum_{i,j} \frac{\partial s_{(0,0)}^{\ell}[T]}{\partial s_{(i,j)}^{\ell-1}[T-\tau]}$. We consider $\sum_{i,j} \frac{\partial \mathcal{L}}{\partial \mathbf{s}_{(i,j)}^{\ell-1}[T-\tau]}$. By applying the chain rule:

$$\sum_{i,j} \frac{\partial \mathcal{L}}{\partial \mathbf{s}_{(i,j)}^{\ell-1}[T-\tau]} = \sum_{i,j} \sum_{\hat{i},\hat{j}} \frac{\partial \mathcal{L}}{\partial \mathbf{s}_{(\hat{i},\hat{j})}^{\ell}[T]} \frac{\partial \mathbf{s}_{(\hat{i},\hat{j})}^{\ell}[T]}{\partial \mathbf{s}_{(i,j)}^{\ell-1}[T-\tau]} \tag{20}$$

When the following conditions are met:

$$\forall \hat{i}, \hat{j}, \frac{\partial \mathcal{L}}{\partial \mathbf{s}_{(\hat{i},\hat{j})}^{\ell}[T]} = 1 \tag{21}$$

We can simplify:

$$\sum_{i,j} \frac{\partial \mathcal{L}}{\partial s_{(i,j)}^{\ell-1}[T-\tau]} = \sum_{\hat{i},\hat{j}} \sum_{i,j} \frac{\partial s_{(\hat{i},\hat{j})}^{\ell}[T]}{\partial s_{(i,j)}^{\ell-1}[T-\tau]} \tag{22}$$

The temporal ERF at delay $\tau$ can thus be calculated by summing the gradients of the loss with respect to all spatial positions at timestep $T - \tau$. □

# B   Details in Detection and Segmentation Experiments

On ImageNet-1K pretraining, we employ three scales of Meta-SDT with three different channel mixer design ((1): Conv-Mixer; (2): MLPixer; (3): SRB) in Table 4 and utilize the hyper-parameters in Table 5 to pre-train models in our paper for further fine-tuning on COCO 2017 and ADE20K datasets. Note that $\epsilon$ represents the channel expand ratio (CHW $\rightarrow \epsilon$ CHW $\rightarrow$ CHW).

For COCO 2017 dataset, We utilize the MMDetection [72] framework to implement the existing models and our method. The object detection and instance segmentation framework strictly follows Mask R-CNN, with a training schedule of $1\times$ (12 epochs). We use a total batch size of 4/GPU, utilize the AdamW optimizer with a learning rate of $1 \times 10^{-4}$ and a weight decay of 0.05. Images are resized and cropped into 1333 $\times$ 800 for training and testing and maintain the ratio. Random horizontal flipping and resize with a ratio of 0.5 was applied for augmentation during training. This pre-training fine-tuning method is a commonly used strategy in ANNs.

For ADE20K dataset, we utilize the MMSegmentation [73] framework. The training configuration strictly encompasses for 160,000 iterations. The batch size is set to 4/GPU, and the AdamW optimizer is used. The learning rate and weight decay parameters are tuned to $2 \times 10^{-4}$ and 0.05, respectively. To speed up training, we warm up the model for 1.5k iterations with a linear decay schedule. All the experiments are conducted on 4 NVIDIA-A100 80GB GPUs.

Table 4: Configurations of different Meta-SDT Variants.

| Stage | # Tokens | Layer Specification | | | Tiny | Medium | Base |
|---|---|---|---|---|---|---|---|
| 1 | $\frac{H}{2} \times \frac{W}{2}$ | Downsampling | | Conv | 7x7 stride 2 | | |
| | | | | Dim | 16 | 24 | 32 |
| | | Conv-based SNN block | SepConv | DWConv | 7x7 stride 1 | | |
| | | | | MLP ratio | 2 | | |
| | | | Channel Mixer | (1)Conv+Conv | $\epsilon = 4$ | | |
| | | | | (2)MLP+MLP | $\epsilon = 4/6$ | | |
| | | | | (3)MLP+Conv | $\epsilon = 4$ | | |
| | $\frac{H}{4} \times \frac{W}{4}$ | Downsampling | | Conv | 3x3 stride 2 | | |
| | | | | Dim | 32 | 48 | 64 |
| | | Conv-based SNN block | SepConv | DWConv | 7x7 stride 1 | | |
| | | | | MLP ratio | 2 | | |
| | | | Channel Mixer | (1)Conv+Conv | $\epsilon = 4$ | | |
| | | | | (2)MLP+MLP | $\epsilon = 4/6$ | | |
| | | | | (3)MLP+Conv | $\epsilon = 4$ | | |
| 2 | $\frac{H}{8} \times \frac{W}{8}$ | Downsampling | | Conv | 3x3 stride 2 | | |
| | | | | Dim | 64 | 96 | 128 |
| | | Conv-based SNN block | SepConv | DWConv | 7x7 stride 1 | | |
| | | | | MLP ratio | 2 | | |
| | | | Channel Mixer | (1)Conv+Conv | $\epsilon = 4$ | | |
| | | | | (2)MLP+MLP | $\epsilon = 4/6$ | | |
| | | | | (3)MLP+Conv | $\epsilon = 4$ | | |
| | | | # Blocks | | 2 | | |
| 3 | $\frac{H}{16} \times \frac{W}{16}$ | Downsampling | | Conv | 3x3 stride 2 | | |
| | | | | Dim | 128 | 192 | 256 |
| | | Transformer-based SNN block | SDSA | RepConv | 3x3 stride 1 | | |
| | | | Channel MLP | MLP ratio | 4 | | |
| | | | # Blocks | | 6 | | |
| 4 | $\frac{H}{16} \times \frac{W}{16}$ | Downsampling | | Conv | 3x3 stride 1 | | |
| | | | | Dim | 192 | 240 | 360 |
| | | Transformer-based SNN block | SDSA | RepConv | 3x3 stride 1 | | |
| | | | Channel MLP | MLP ratio | 4 | | |
| | | | # Blocks | | 2 | | |

Table 5: Hyper-parameters for pre-training on ImageNet-1K

| Hyper-parameter | Settings | Hyper-parameter | Settings |
|---|---|---|---|
| Model size | T/M/B | Timestemp | 4 |
| Epochs | 200 | Resolution | 224*224 |
| Batch size | 1568 | Optimizer | LAMB |
| Base learning rate | 6e-4 | Learning rate decay | Cosine |
| Warmup eopchs | 10 | Weight decay | 0.05 |
| Random augment | 9/0.5 | Mixup | None |
| Cutmix | None | Label smoothing | 0.1 |

# C Limitations

This work presents several avenues for future exploration, such as how neuronal dynamics parameters influence ST-ERF in more dynamic and diverse SNNs. Given that one of SNN's major successes stems from its inherent membrane potential memory update mechanism, this represents a particularly worthwhile direction for deeper investigation. We will further explore the interactions between spiking neurons' neurodynamics and the networks' temporal response in the future. Nevertheless, this work provides a viable analytical framework for understanding SNN model behavior, with practical implications for architectural design across various levels of SNNs.

