# OpenReview forum: "Unveiling the Spatial-temporal Effective Receptive Fields of Spiking Neural Networks"
_NeurIPS.cc/2025/Conference — NeurIPS 2025 poster_

### Official Review · Reviewer_X2Er · 2025-06-26

**Clarity:** 2
**Significance:** 2
**Originality:** 2
**Rating:** 4
**Confidence:** 5

**Summary:**

This paper introduces the Spatio-Temporal Effective Receptive Field (ST-ERF), a gradient-based metric for analyzing how spiking neural networks (SNNs) process spatial and temporal information. Using ST-ERF, the authors evaluate several existing Spiking Vision Transformer (S-ViT) models and identify a limitation in their global receptive field coverage. To address this, they propose two new channel-mixer designs—MLPixer and Splash-and-Reconstruct (SR) Block—which aim to enhance global spatial ERF in early stages. The proposed methods are evaluated on COCO 2017 and ADE20K, showing improvements in detection and segmentation tasks.

**Questions:**

See Weaknesses

**Ethical Concerns:**

["NO or VERY MINOR ethics concerns only"]

**Final Justification:**

The authors have addressed my concerns clearly and conducted extensive additional experiments to support their claims. I appreciate the clarification of the architectural comparisons, the added analysis of spatial ERFs across various ViT variants, and the new temporal benchmark results. The connection between ERF diagnosis and the proposed module designs is now better explained. Overall, the rebuttal strengthens the paper, and I have decided to raise my score.

**Quality:**

2

**Strengths And Weaknesses:**

Strengths:
1. The paper proposes ST-ERF, a well-motivated and theoretically grounded framework to understand the spatial and temporal behavior of SNNs.
2. The diagnostic application of ST-ERF to Transformer-based SNNs is insightful and helps identify architectural bottlenecks related to global receptive field coverage.
3. The proposed MLPixer and SR Block modules are lightweight and show performance improvements on standard vision benchmarks.
Weaknesses:
1.  The comparison between ViT-B (a lightweight, pure Transformer) and S-ViTs (which include heavy neuromorphic modules like SPS blocks) is not well controlled. Differences in ERF may arise from structural depth and convolutional paths rather than the spiking mechanism itself.
2. ERF distributions can vary with activation types. No comparisons are made with ReLU/GELU or quantized ViT variants.  Modern ViT variants (e.g., Swin-T, PVT) are excluded, weakening claims about S-ViT inferiority in global modeling.
3. The paper does not isolate the effects of MLPixer and SR Block via controlled replacements or report their individual contributions. Though the method is motivated by temporal modeling, only static image datasets (COCO, ADE20K) are used. No temporal benchmarks are evaluated.
4. The path from ERF diagnosis to MLPixer/SR block design lacks clarity. More intuition, derivation, or visualization is needed to justify the structural choices.

---

> ### Author Rebuttal · Authors · 2025-07-29
>
> > Q1: Concerns about the comparison between ViT-B and S-ViTs:
>
> A1: We sincerely appreciate your attention to the clarity of Figure 2 and the ensuing concerns it raised regarding the rigor of our experimental controls. We understand your concerns and would like to clarify the original intent behind Figure 2 and its key findings.
>
> The primary purpose of Figure 2 was to: provide a visualization and comparison of the Spatial Effective Receptive Fields (*Spatial ERFs*) for comparable ANN- and SNN-based Transformer models. This approach was designed to profoundly reveal the underlying learning mechanisms and behavioral patterns. Our experimental design rigorously controlled for architectural consistency:
> 1.  **ViT-like Architecture Comparison Group**: The Vanilla ViT (ANN) and Spikformer/SDT-V1 (SNN) exhibit identical module structure, with the core difference residing solely in the activation function (ANN: ReLU; SNN: Spiking).
> 2.  **Metaformer Architecture Comparison Group**: The CAFormer-s18 (ANN, Metaformer paradigm) and its counterpart Meta-SDT-S (SNN) are precisely matched in critical design elements such as convolutional pathways and mixed hierarchical block design.
>
> The experimental results clearly demonstrate that, under strict control of architectural variables, the ANN and SNN models exhibit strikingly similar Spatial ERF distribution characteristics:
> *   ViT-like architecture's ERF manifests a `checkerboard pattern`, which is highly similar to the ERF patterns observed in both Spikformer (8-512) and SDT-V1.
> *   Similarly, the ERF of both CAFormer-s18 (ANN) and Meta-SDT-S (SNN) display a highly `localized pattern`, remarkably analogous to that of traditional CNN networks. This indicates the model's focus is primarily concentrated on local regions, with relatively limited interaction occurring with other image patches.
>
> Therefore, the consistent ANN/SNN ERF patterns presented in Figure 2 precisely serve as compelling evidence for the effectiveness of our experimental design and the rigor of our variable control (particularly concerning architecture). This observed consistency directly captures the key phenomenon we aimed to introduced: Different architectures influence the learning behavior of ANN vs. SNN models, which is the foundation of our investigation on the limitation in spatial ERF distribution in current S-ViT models.
>
> To present the aforementioned experimental design and core findings more clearly, we will undertake a comprehensive reorganization of Figure 2:
> *   ANN models will be presented side-by-side with their structurally identical SNN counterparts.
> *   Move QKFormer (10-384) visualization to Appendix, as its architecture is not quite comparable to the other models.
>
> > Q2: More Spatial ERF visualizations for other ViT models:
>
> We have supplemented the following ViT variants to provide a more comprehensive presentation of the Spatial ERF characteristics across different ViT models.
> Since the rebuttal contents are not allowed to contain image links, we will give an overall comparison in the following table:
>
> | Model                     |ImageNet Acc-1(%)  | Spatial ERF Pattern       | Interaction Intensity with other patches |
> |---------------            |---------------    |---------------------      |--------------------|
> | Meta-SDT-S(SNN)           | 75.3              | Conv-like Gaussian facula              | +            |
> | ViT-Tiny (ReLU) (ANN)            | 73.0              | Checkerboard              | ++            |
> | ViT-Tiny (GELU) (ANN)           | 75.4              | Checkerboard              | ++            |
> | Q-ViT DeiT-Tiny (4-bit) (ANN)   | 74.3              | Checkerboard              | ++           |
> | Swin-Tiny (ANN)                 | 80.9              | Hierarchical block        | +++           |
> | PVTv2-B0 (ANN)                  | 70.5              | Conv-like Periodic facula | ++            |
>
> Note that all the above ANN-based models exhibit a more global receptive field distribution compared to the S-ViT models, which is consistent with the findings in our paper. The interaction intensity with other patches in ANN-based models is also higher, indicating a more global receptive field distribution.
> We will include these visual aids into the revised manuscript to facilitate a more comprehensive comparison of the Spatial ERF distributions across different ViT models.
>
> > Q3: Concerns about the controlled replacements of MLPixer and SR Block and unclear analysis path from spatial ERF to MLPixer/SR block design:
>
> A3.1: We thank the reviewer for raising this point. In our detection and segmentation experiments, all blocks in Stages 1 & 2 were systematically replaced with MLPixer or SR Block, ensuring that block type was the sole variable. This controlled setup allowed us to isolate the impact of architectural design on model performance.
>
> A3.2: We acknowledge that the connection between Spatial ERF analysis and our block design choices was not fully articulated. The key insight is that the Spatial ERF distribution of the S-ViTs, as visualized in Figure 2, reveals a more localized pattern: having a stronger focus on local features but limited global context. This suggests that current S-ViTs may not fully leverage the global receptive field capabilities of their ANN counterparts. Throughout the exploration, we observed that the MLPixer and SR Block designs, which are more suited for global feature extraction, align better with the Spatial ERF distribution of S-ViTs. This is why we chose to replace the original blocks with MLPixer and SR Block in our experiments. Our experiments demonstrated that these replacements led to improved performance in detection and segmentation tasks, further supporting the hypothesis that S-ViTs can benefit from architectural designs that enhance global context understanding.
>
> > Q4: Question on the benchmarking of S-ViTs: Provide more temporal benchmarks:
>
> We appreciate your suggestion to provide more temporal benchmarks for S-ViTs. We have conducted additional experiments on the DVS128 Gesture datasets based on the QKFormer network, which are commonly used benchmarks for event-based vision classification tasks.
>
> The baseline network is QKFormer-PE1(4-layer-res)-S1-PE2(2-layer-res)-S2, containing two stages of Q-K attention (S1 & S2) and the corresponding positional encoding blocks(PE1 & PE2). We apply the block design of MLPixer and SR block on the PEs. Note that because the original QKFormer has residual paths in PEs, we also added the block design of MLPixer and SR block on the residual paths to see their performance
>
> The results are summarized in the following table:
>
> | Architecture                          |Step | Param (M) | DVS128Ges. Acc. (%) |
> |--------------------------------------|----|----|-----|
> | **QKFormer-PE1-S1-PE2-S2**           |$16$|1.99|97.9|
> | ✓ ***+PE1-MLPixer ($\epsilon=2$)***  |$16$|1.91|97.9|
> | ✓ ***+PE1-respath-MLPixer***              |$16$|1.91|97.2|
> | ♠ ***+PE1-MLPixer ($\epsilon=3$)***  |$16$|1.93|98.3|
> | ♠ ***+PE1-respath-MLPixer***              |$16$|1.93|97.2|
> | ► ***+PE1-SR ($\epsilon=2$)***       |$16$|1.92|98.3|
> | ► ***+PE1-respath-MLPixer***              |$16$|1.92|98.6|
> | ✓ ***+PE2-MLPixer ($\epsilon=2$)***  |$16$|1.21|97.6|
> | ✓ ***+PE2-respath-MLPixer***              |$16$|1.21|98.3|
> | ♠ ***+PE2-MLPixer ($\epsilon=3$)***  |$16$|1.26|97.9|
> | ♠ ***+PE2-respath-MLPixer***              |$16$|1.26|98.3|
> | ► ***+PE2-SR ($\epsilon=2$)***       |$16$|1.48|98.3|
> | ► ***+PE2-respath-MLPixer***              |$16$|1.48|98.6|
>
> Exp Setting:
> | **Hyperparameter**        | **Settings for DVS128 Gesture**   |
> |-------------------------  |-------------|
> | `batch_size`              | 16          |
> | `T` (time steps)          | 16          |
> | `opt`                     | adamw       |
> | `opt_eps`                 | 1e-08       |
> | `weight_decay`            | 0.06        |
> | `momentum`                | 0.9         |
> | `sched` (scheduler)       | cosine      |
> | `lr`                      | 0.001       |
> | `warmup_lr`               | 1e-05       |
> | `min_lr`                  | 2e-05       |
> | `epochs`                  | 192         |
> | `decay_epochs`            | 20          |
> | `warmup_epochs`           | 10          |
> | `decay_rate`              | 0.1         |
> | `smoothing` (label smooth)| 0.1         |
> | `mixup`                   | 0.5         |
> | `cutmix`                  | 0.0         |
> | `cutmix_minmax`           | `None`      |
> | `mixup_prob`              | 0.5         |
> | `mixup_switch_prob`       | 0.5         |
>
> We also have conducted additional experiments on the Event-based tracking tasks on FE108 dataset and VisEvent dataset. The results are summarized in the following tables:
>
> | Model         |#T|#P| FE108 Tracking (AUC/PR) | VisEvent Tracking (AUC/PR) |
> |:--------------|--|--|:-------------------------|:----------------------------|
> | SD-Track(Tiny)|4*1|19.61|56.7/89.1|35.4/48.7|
> | SD-Track(Tiny)|1*4|19.61|59.0/91.3|35.6/49.2|
> | SD-Track(Tiny)--MLPixer ($\epsilon=4$)|1*4|20.21|57.1/89.2|33.7/47.3|
> | SD-Track(Tiny)--MLPixer ($\epsilon=6$)|1*4|22.99|57.9/90.1|34.5/48.9|
> | SD-Track(Tiny)--SR ($\epsilon=4$)|1*4|21.43|58.2/88.5|33.8/48.0|
>
> Exp Setting:
> | **Hyperparameter**        | **Settings for FE108** | **Settings for VisEvent** |
> |-------------------------  |-------------|-------------|
> | `epochs`                  | 100         | 100         |
> | `opt`                     | adamw       | adamw       |
> | `sample number`           | 60000       | 30000       |
> | `lr`                      | 4e-4        | 4e-4        |
> | `decay_epochs`            | 80          | 80          |
>
> These results confirm that the MLPixer and SR block designs preserve S-ViT performance. While current temporal benchmarks show promise, further improvement is possible. We appreciate your insightful suggestion, which has highlighted this important direction for exploration.

---

> > ### Comment · Reviewer_X2Er · 2025-08-05
> >
> > Thank you for the detailed and thoughtful response. The authors have addressed my concerns clearly and conducted extensive additional experiments to support their claims. I appreciate the clarification of the architectural comparisons, the added analysis of spatial ERFs across various ViT variants, and the new temporal benchmark results. The connection between ERF diagnosis and the proposed module designs is now better explained. Overall, the rebuttal strengthens the paper, and I have decided to raise my score.

---

> > > ### Author Response · Authors · 2025-08-05
> > > **Response to Reviewer X2Er**
> > >
> > > Thank you very much for your thoughtful and encouraging feedback, and for raising your score. We are particularly grateful that you recognized our efforts on the additional experiments and the detailed clarifications. We will ensure that the improved explanations on architectural comparisons, the ERF analysis, the new benchmark results, and the connection between diagnosis and design are all carefully integrated into the final version of the manuscript to reflect the strengthened claims.

---

### Official Review · Reviewer_oAHP · 2025-06-30

**Clarity:** 3
**Significance:** 4
**Originality:** 3
**Rating:** 5
**Confidence:** 5

**Summary:**

This paper introduces the Spatio-Temporal Effective Receptive Field (ST-ERF), an extension of the traditional Effective Receptive Field concept from ANNs to SNNs that considers both spatial and temporal dynamics, and shows that ST-ERFs are Gaussian distributed in space and decay over time according to a power law. Based on ST-ERF, this paper also reveals that existing transformer-based SNNs suffer from overly localized receptive fields that hinder long-sequence modeling. The authors attribute this issue to the design of channel mixers and propose two novel modules, MLPixer and Splash-and-Reconstruct Block, to enhance early global receptive fields. Experiments confirm the effectiveness of these modules. In summary, this paper defines ST-ERF for SNNs, identifies a key limitation in transformer-based architectures, and offers effective mixer designs to address it.

**Questions:**

1. In Figure 6, the receptive field of SR-SDT is mainly concentrated in the lower-left region. What causes this localization?

2. More experiments should conducted to compare to SOTA  methods?

3. The authors have conducted comprehensively analysis about ERF in SNNs,  how can we adapt these conclusions to practical applications?

**Ethical Concerns:**

["NO or VERY MINOR ethics concerns only"]

**Final Justification:**

The weaknesses and questions I mentioned have been well addressed by the author.

**Limitations:**

Yes

**Quality:**

3

**Strengths And Weaknesses:**

Strengths:

1.This work introduces the concept of the Effective Receptive Field to the SNN community, filling a critical gap and uncovering the key properties of ST-ERF.

2.By analyzing the spatio-temporal effective receptive field (ST-ERF) of transformer-based SNNs, this paper identifies their limitations and proposes novel modules to address them. The proposed modules are well motivated.

Weaknesses:

1.In the Verification of Gaussian-like distribution of ERF, the paper should also include the ERF of the more commonly used LIF model to check if it likewise follows a Gaussian distribution.

2.There is no comparison with other state-of-the-art methods.

---

> ### Author Rebuttal · Authors · 2025-07-29
>
> > Q1: ST-ERF in commonly used LIF model :
>
> A1: We apologize for the oversight in not including the ST-ERF visualization for the commonly used LIF model in our original submission. We have now finished this visualization. The detailed explanation of the ST-ERF for the LIF model is shown below the table since the rebuttal contents are not allowed to contain image links. We will include this visualization in the appendix to provide a more comprehensive understanding of the ST-ERF in commonly used LIF models. (**SG**: surrogate gradient function)
>
>
> | Model         | Spatial ERF Pattern |horizontal $\sigma^2$|vertical $\sigma^2$|
> |---------------|---------------------|-----|----------|
> | I-LIF         | 2D Gaussian-like    | 4.1 | 4.1 |
> | NI-LIF        | 2D Gaussian-like    | 3.8 | 3.8 |
> | **LIF(SG=$\mathbf{arctan(\alpha=2)}$)**| 2D Gaussian-like    | 1.3 | 1.3 |
> | **LIF(SG=$\mathbf{arctan(\alpha=4)}$)**| 2D Gaussian-like    | 1.3 | 1.3 |
> | **LIF(SG=$\mathbf{arctan(\alpha=8)}$)**| 2D Gaussian-like    | 1.3 | 1.3 |
> | **LIF(SG=$\mathbf{Sigmoid(\alpha=2)}$)**| 2D Gaussian-like    | 1.6 | 1.6 |
> | **LIF(SG=$\mathbf{Sigmoid(\alpha=4)}$)**| 2D Gaussian-like    | 1.6 | 1.6 |
> | **LIF(SG=$\mathbf{Sigmoid(\alpha=8)}$)**| 2D Gaussian-like    | 1.6 | 1.6 |
>
> Experiment Setting:
> | **Hyperparameter**        | **Value**   |
> |-------------------------  |-------------|
> | `arch.`                   | Spiking-CNN |
> | `layers`                  | 20          |
> | `trials of running`       | 50          |
>
> The LIF-based ST-ERF visualization results show that the ST-ERF distribution follows the 2D Gaussian-like pattern, which is consistent with the findings in our paper. The horizontal and vertical $\sigma^2$ values are smaller than those of the I-LIF and NI-LIF models, indicating a more localized receptive field distribution.
>
> > Q2: More comparison to SOTA models on various tasks:
>
> A2: We appreciate your suggestion. We have conducted additional experiments on event-based tracking tasks using the FE108 and VisEvent datasets to compare with SOTA models.
>
> **Event-based Tracking Experiments**: To comprehensively evaluate our ERF-guided architectural improvements across different temporal processing domains, we extended our evaluation to event-based object tracking tasks. Event-based tracking presents unique challenges as it requires processing asynchronous, sparse event streams with high temporal resolution while maintaining spatial consistency across time. Unlike traditional RGB tracking, event cameras capture brightness changes asynchronously, producing sparse event streams that require specialized temporal aggregation and spatial attention mechanisms.
>
> Our experimental setup employed the SD-Track architecture as the baseline, which integrates event-based feature extraction specifically designed for processing event streams. We systematically replaced the core processing blocks with our proposed MLPixer and SR block variants while maintaining identical experiment setups. The FE108 dataset provides high-speed tracking scenarios with rapid object motion, while VisEvent offers more diverse tracking conditions with varying lighting and occlusion patterns. The results are summarized in the following tables:
>
> | Model         |#T|#P| FE108 Tracking (AUC/PR) | VisEvent Tracking (AUC/PR) |
> |:--------------|--|--|:-------------------------|:---------------------------|
> | ODTrack       |1*1|92.83|43.2/69.7|24.7/34.7|
> | HIPTrack     |1*1|120.51|50.8/81.0|32.1/45.2|
> | SD-Track(Tiny)|4*1|19.61|56.7/89.1|35.4/48.7|
> | SD-Track(Tiny)|1*4|19.61|59.0/91.3|35.6/49.2|
> | SD-Track(Tiny)-MLPixer ($\epsilon=4$)|1*4|20.21|57.1/89.2|33.7/47.3|
> | SD-Track(Tiny)-MLPixer ($\epsilon=6$)|1*4|22.99|57.9/90.1|34.5/48.9|
> | SD-Track(Tiny)-SR ($\epsilon=4$)|1*4|21.43|58.2/88.5|33.8/48.0|
>
> The results demonstrate that our architectural modifications not only preserve but in some cases improve tracking performance across both precision (PR) and area-under-curve (AUC) metrics. Notably, MLPixer with $\epsilon=6$ and SR blocks show consistent improvements, validating that our ERF-guided architectural choices are effective across diverse temporal processing tasks beyond static image classification.
>
> > Q3: In Figure 6, the receptive field of SR-SDT is mainly concentrated in the lower-left region. What causes this localization?
>
> A3: The localization of the receptive field in the lower-left region of Figure 6 is primarily due to the model's learning selectivity. This phenomenon occurs because the analysis framework is based on the propagation of gradients, which can lead to a shift in the effective receptive field when the gradient stimuli backwards towards the whole network. The input data also plays a role in this localization effect, as it influences how the model learns to focus on specific regions of the input space.
>
> To verify whether the Spatial ERF is pixel-wise invariant, we can conduct experiments by shifting the input gradient stimuli and observing the corresponding changes in the spatial ERF pattern. If we use a stimuli that simply shifts the focus by a certain amount, we would see a totally different receptive field distribution. This result demonstrates that in a properly-trained network, every single pixel has its own unique ERF pattern. To verify whether the training data is responsible for this effect, we can conduct ablation studies by sampling a subset of the original training data and observing the changes in the spatial ERF given the same gradient stimuli. We will include the visualization explanation in the updated manuscript to clarify this point further. The comparison is below the table since the rebuttal contents are not allowed to contain image links.
>
> |Model|Gradient Stimuli|Spatial ERF Pattern in Stage 3|
> |------------|--------|--------------|
> | SDT-v3-Tiny|Centered|Conv-like Gaussian facula|
> | SDT-v3-Tiny|Shifted $(+5)_{vertical}$ pixels|Conv-like Gaussian facula (shifted $(+5)_{vertical}$)|
> | SDT-v3-Tiny|Shifted $(+5)_{horizontal}$ pixels|Conv-like Gaussian facula (shifted $(+5)_{horizontal}$)|
> | SDT-v3-Tiny-MLPixer($\epsilon=4$)|Centered|Patchy distribution with regional selectivity|
> | SDT-v3-Tiny-MLPixer($\epsilon=4$)|Shifted $(+5)_{vertical}$ pixels|Patchy distribution with regional selectivity (totally different from original)|
> | SDT-v3-Tiny-MLPixer($\epsilon=4$)|Shifted $(+5)_{horizontal}$ pixels|Patchy distribution with regional selectivity (totally different from original)|
> | SDT-v3-Tiny-SR($\epsilon=4$)|Centered|Patchy distribution with regional selectivity|
> | SDT-v3-Tiny-SR($\epsilon=4$)|Shifted $(+5)_{vertical}$ pixels|Patchy distribution with regional selectivity (totally different from original)|
> | SDT-v3-Tiny-SR($\epsilon=4$)|Shifted $(+5)_{horizontal}$ pixels|Patchy distribution with regional selectivity (totally different from original)|
>
> The exploration of the spatial ERF in S-ViTs, particularly in the context of MLPixer and SR blocks, reveals that the design of these architectures plays a crucial role in introducing a disaggregated-then-reintegrated receptive field structure. This finding highlights the importance of architectural choices in shaping the effective receptive fields and their localization properties.
>
> > Q4: How to adapt ERF-based conclusions in SNNs to practical applications?
>
> A4: Through the comprehensive ST-ERF analytical framework, we demonstrate that SNN blocks with specialized designs (including but not limited to task-specific convolutional operators, adaptive/windmill convolutions, or attention-enhanced mechanisms such as self-attention/SE modules) exhibit distinct input feature preferences. These spatially modulated preferences facilitate the interpretability of block-level behaviors across different tasks, thereby providing principled guidance for: (1) designing more powerful and computationally efficient task-specific architectures, or (2) formulating constraints for Neural Architecture Search (NAS) to discover optimal topologies.
>
> Temporally, our analysis reveals dynamic preference variations across timesteps in SNN implementations (under current LIF/I-LIF activation paradigms). The intrinsic membrane potential updating mechanism inherently strengthens responses to recent inputs while attenuating distant temporal information—a property we term temporal locality bias. This finding motivates further investigation via the ST-ERF framework to quantify how neuronal dynamics parameters govern spatiotemporal receptive fields in more biologically plausible SNNs. Such exploration could yield fundamental insights into the shared principles and divergences between brain-inspired networks and biological neural systems.

---

> > ### Comment · Reviewer_oAHP · 2025-08-05
> > **Official Comment by Reviewer oAHP**
> >
> > Thanks for your response. The rebuttal addresses my concern. I will increase my rating to 5.

---

> > > ### Author Response · Authors · 2025-08-05
> > > **Response to Reviewer oAHP**
> > >
> > > Thank you very much for your thoughtful feedback and recognition of our work. We are pleased that our response addressed your concerns, and we are grateful for the score increase. We will ensure that the points discussed are integrated into the revised manuscript to further improve its quality.

---

### Official Review · Reviewer_Yc3k · 2025-06-30

**Clarity:** 3
**Significance:** 3
**Originality:** 3
**Rating:** 4
**Confidence:** 5

**Summary:**

This paper introduces the Spatio-Temporal Effective Receptive Field (ST-ERF), a novel framework for analyzing learning behaviors in Spiking Neural Networks (SNNs). Extending traditional receptive field analysis to account for SNNs’ temporal dynamics, the authors establish two fundamental properties of ST-ERF: **Spatial Gaussianity**: ST-ERF exhibits Gaussian-like distributions in the spatial domain. **Temporal Power-law Attenuation**: Influence decays following a power-law pattern over time, driven by neuronal dynamics. Applying ST-ERF to analyze Transformer-based SNNs, the paper reveals that current architectures suffer from limited global receptive fields, impairing performance on visual long-sequence tasks. Crucially, the authors identify the channel-mixer design as the primary bottleneck. By proposing two novel block design: **MLPixer** and **SR block**, these designs significantly improve spatial ERF coverage in early network stages, enabling state-of-the-art performance on COCO and ADE20K benchmarks. The work bridges theoretical understanding (ST-ERF properties) and practical advancements (new architectures) for SNNs in complex vision tasks.

**Questions:**

1. Is there any further connections between ST-ERF and other biological theories of neural dynamics? The paper mentions that ST-ERF is a theoretical framework, but it does not discuss how it relates to existing biological theories of neural dynamics. Are there any connections between ST-ERF and other theories, such as the theory of synaptic plasticity or the theory of neural oscillations?
2. Proposition 1 correlates spatial ERF with ANN-like behavior under the condition of "pure instantaneous response". However, many SNNs use time-coding (e.g., delayed coding), which defeats this premise. How is the ST-ERF compatible with the scheme of encoding information in pulse timing?

**Ethical Concerns:**

["NO or VERY MINOR ethics concerns only"]

**Final Justification:**

This paper introduces a solid theoretical framework (ST-ERF) for understanding SNN spatiotemporal dynamics, addressing an important research gap. The mathematical foundations are sound, and the identification of receptive field limitations in existing architectures leads to practical improvements with demonstrated performance gains.

While the evaluation scope could be broader, the work makes meaningful contributions to both SNN theory and practice. The technical quality is adequate and the proposed methods are well-validated. Overall, the strengths outweigh the weaknesses.

**Limitations:**

Yes. As discussed in the appendix, the authors transparently acknowledge the technical constraints (e.g., unexplored neurodynamics influences on ST-ERF, narrow experimental validation, and its connection with scalability). The authors also provide a clear roadmap for future work: exploring ST-ERF in more complex conditions.

**Paper Formatting Concerns:**

No major formatting issues were identified in this paper.

**Quality:**

3

**Strengths And Weaknesses:**

**Strengths:**
1. The paper provides convincing proofs on one proposition and two properties of ST-ERF, establishing a solid theoretical basis for understanding SNN learning dynamics and representing the first formal adaptation of ERF theory to SNN spatiotemporal dynamics.

2. The proposed MLPixer/SR Blocks demonstrate consistent performance gains across all metrics, with improvements up to +3.6 mIoU, providing solid validation of the theoretical insights.

3. Intuitive visualizations including ERF heatmaps (Figures 2, 4, 6) and decay curves (Figure 5) effectively illustrate spatial-temporal properties and make complex concepts tangible.

4. Original analysis revealing channel-mixer design as the global ERF bottleneck, with the SR Block's MLP-convolution hybrid offering a novel solution for SNN receptive field enhancement.

5. Thoughtful adaptation of architectural advances (Metaformer/MLPixer) to overcome SNN limitations, demonstrating how cross-paradigm synthesis can drive meaningful progress in the field.

**Weaknesses:**
1. Some figures contain ambiguities, such as Figure 3 where the second block design appears to lack residual connections, requiring verification and clearer explanations.

2. Figure 5(b) legend lacks clear explanation of which experiments (a)-(d) each element represents, hindering interpretation.

3. The paper primarily focuses on visual long-sequence tasks, which may limit the generalizability of findings to other domains and applications, suggesting need for broader task validation to demonstrate framework versatility.

---

> ### Author Rebuttal · Authors · 2025-07-29
>
> > Q1: Corrigendum in Figure 3 where the second block design appears to lack residual connections.
>
> A1: We sincerely appreciate your meticulous review and valuable feedback. You rightly identified the missing residual connection in Figure 3, which was indeed an oversight in our original manuscript. We sincerely apologize for this error.
>
> We have carefully revised Figure 3 in the updated manuscript—the residual pathway is now explicitly illustrated in the MLPixer block (black solid arrows). Furthermore, our actual implementation (Code in supplementary materials: `erf_compute/spatial_Erf/erf_sdt/models/sdtv3.py`, Lines 357 & 395) and mathematical formulations in the original manuscript (Eq. (12) & (13): $\mathbf{X}''= \mathbf{X}'+\mathrm{Mixer}_{ch}(\mathbf{X}')$) confirm that the residual connection was consistently incorporated in the design.
>
> > Q2: Figure 5(b) legend lacks clear explanation of which experiments (a)-(d) each element represents, hindering interpretation.
>
> A2: We apologize for the lack of clarity in the legend of Figure 5(b). In Figure 5(b), we aimed to illustrate the temporal ERF patterns of Spiking CNNs with different spike activations. We employ a 20-layer deep SCNN network to derive temporal ERFs across four distinct surrogate functions. From subfigures (a) to (d), the surrogate functions are as follows: (a): $\mathrm{arctan(\cdot)}$, (b): $\mathrm{Sigmoid}(\cdot)$, (c): $\mathrm{Rect(\cdot)}$, (d): $\mathrm{Poly(\cdot)}$.
>
> In each subfigure, we visualize the temporal ERF patterns at different decay factors. Considering that the spike activations are achieved by LIF neurons, we can describe the forward process as follows:
> $$
> \begin{aligned}
>     \mathbf{v}^{\ell}[t]&=\mathbf{h}^{\ell}[t-1]+f({\mathbf{w}^{\ell}},\mathbf{x}^{\ell-1}[t-1]), &\text{(Charging function)} \\
>     \mathbf{s}^{\ell}[t]&=\mathbf{\Theta}(\mathbf{v}^{\ell}[t]-\vartheta), &\text{(Firing function)}\\
>     \mathbf{h}^{\ell}[t]&=
>         \beta\mathbf{v}^{\ell}[t]- \vartheta \mathbf{s}^{\ell}[t], &\text{soft reset}
>     \end{aligned}
> $$
>
> Some amount of works define the LIF model as
>
> $$
> \begin{aligned}
>     \mathbf{h}^{\ell}[t]&=
>         (1-\frac{1}{\tau})\mathbf{v}^{\ell}[t]- \vartheta \mathbf{s}^{\ell}[t], &\text{soft reset}
>     \end{aligned}
> $$
> where $\tau$ is the decay factor. However, in our theoretical framework, we use $\beta=1-\frac{1}{\tau}$ for convenience while ensuring consistent monotonicity between $\beta$ and $\tau$
>
> Proven by Appendix D of the original manuscript, the temporal ERF is the temporal effective receptive field decays exponentially with time delay $\tau$, and the decay rate is primarily determined by the membrane potential decay constant $\beta$, i.e., $\underset{\mathcal{T}}{\mathrm{ERF}}(\cdot,\tau) \propto \beta^{\tau}$. The four experiments used decay factors of 2.0, 1.33, 1.2, and 1.14, respectively, which may not be the commonly-said decay factors in many essays. Throughout the experiments, we observed that the fitted curves closely followed the trends of the original data, validating our theoretical framework.
>
> > Q3: Broader task validation to demonstrate framework versatility.
>
> A3: We appreciate your suggestion to validate our framework across a broader range of tasks. In response, we have conducted additional experiments on **(a): DVS data classification tasks** using DVS128Gesture, and **(b): event-based tracking tasks** using the FE108 and VisEvent datasets.
>
> **(a): DVS data classification tasks.** The baseline network is QKFormer-PE1(4-layer-res)-S1-PE2(2-layer-res)-S2, containing two stages of Q-K attention (S1 & S2) and the corresponding positional encoding blocks(PE1 & PE2). We apply the block design of MLPixer and SR block on the PEs. Note that because the original QKFormer has residual paths in PEs, we also added the block design of MLPixer and SR block on the residual paths to see their performance. The results are summarized in the following table:
> | Architecture                          |Step | Param (M) | DVS128Ges. Acc. (%) |
> |--------------------------------------|----|----|-----|
> | **QKFormer-PE1-S1-PE2-S2**           |$16$|1.99|97.9|
> | ✓ ***+PE1-MLPixer ($\epsilon=2$)***  |$16$|1.91|97.9|
> | ✓ ***+PE1res-MLPixer***              |$16$|1.91|97.2|
> | ♠ ***+PE1-MLPixer ($\epsilon=3$)***  |$16$|1.93|98.3|
> | ♠ ***+PE1res-MLPixer***              |$16$|1.93|97.2|
> | ► ***+PE1-SR ($\epsilon=2$)***       |$16$|1.92|98.3|
> | ► ***+PE1res-MLPixer***              |$16$|1.92|98.6|
> | ✓ ***+PE2-MLPixer ($\epsilon=2$)***  |$16$|1.21|97.6|
> | ✓ ***+PE2res-MLPixer***              |$16$|1.21|98.3|
> | ♠ ***+PE2-MLPixer ($\epsilon=3$)***  |$16$|1.26|97.9|
> | ♠ ***+PE2res-MLPixer***              |$16$|1.26|98.3|
> | ► ***+PE2-SR ($\epsilon=2$)***       |$16$|1.48|98.3|
> | ► ***+PE2res-MLPixer***              |$16$|1.48|98.6|
>
> **(b): Event-based tracking tasks.** Our experimental setup employed the SD-Track architecture as the baseline, which integrates event-based feature extraction specifically designed for processing event streams. We systematically replaced the core processing blocks with our proposed MLPixer and SR block variants while maintaining identical experiment setups. The results are summarized in the following tables:
> | Model         |#T|#P| FE108 Tracking (AUC/PR) | VisEvent Tracking (AUC/PR) |
> |:--------------|--|--|:-------------------------|:---------------------------|
> | ODTrack       |1*1|92.83|43.2/69.7|24.7/34.7|
> | HIPTrack     |1*1|120.51|50.8/81.0|32.1/45.2|
> | SD-Track(Tiny)|4*1|19.61|56.7/89.1|35.4/48.7|
> | SD-Track(Tiny)|1*4|19.61|59.0/91.3|35.6/49.2|
> | SD-Track(Tiny)-MLPixer ($\epsilon=4$)|1*4|20.21|57.1/89.2|33.7/47.3|
> | SD-Track(Tiny)-MLPixer ($\epsilon=6$)|1*4|22.99|57.9/90.1|34.5/48.9|
> | SD-Track(Tiny)-SR ($\epsilon=4$)|1*4|21.43|58.2/88.5|33.8/48.0|
>
> The results demonstrate that our architectural modifications not only preserve but in some cases improve tracking performance across both precision (PR) and area-under-curve (AUC) metrics. Notably, MLPixer with $\epsilon=6$ and SR blocks show consistent improvements, validating that our ERF-guided architectural choices are effective across diverse temporal processing tasks beyond static image classification.
>
> > Q4: Further connections between ST-ERF and other biological theories of neural dynamics?
>
> A4.1 **Connections with neural oscillation theory:** While the current ST-ERF framework focuses on temporal-domain assessment of dynamic feature variations, its natural extension to frequency-domain analysis would unveil neuronal response mechanisms across spectral bands. Such a multidimensional spatiotemporal-frequency analytical approach would: (i) deepen understanding of feature extraction processes in biological neural populations, and (ii) advance bio-plausible computational models for next-generation neuromorphic chips (e.g., Loihi), particularly in cognitive tasks demanding sophisticated time-frequency processing (e.g., working memory and spatial navigation). This proposed expansion aligns with the emerging paradigm of hybrid time-frequency analysis in computational neuroscience.
>
> A4.2 **Connections with synaptic plasticity theories:**  Although the current ST-ERF framework is based on gradients, a simple yet potential method to incorporate STDP into the proposed ST-ERF framework to the analysis of synaptic weights can be modifying the core definition in the original manuscript Eq.(3) with an STDP window function $g(\Delta t)$, where $\Delta t = t_{\text{post}} - t_{\text{pre}} $ captures pre-/post-synaptic timing relationships. This allows explicit modeling of how synaptic efficacy modulates input-output sensitivity. The spatial (Eq.(4)) and temporal (Eq.(5)) decompositions of ST-ERF further benefit from STDP-aware formulations: Spatial ERFs should weight contributions by STDP-driven synaptic updates $w(t,\tau) \propto \eta \sum_{\Delta t} g(\Delta t) I_{\text{pre}}(t - \tau) I_{\text{post}}(t)$, while temporal ERFs can directly adopt the bi-exponential form to reflect STDP’s asymmetric time constants. This unified perspective may reveal how STDP dynamically sculpts spatiotemporal processing, which can be a critical insight bridging synaptic mechanisms with system-level information encoding.
>
> > Q5: How is the ST-ERF compatible with the scheme of encoding information in pulse timing?
>
> A5: Our framework employs **dynamic firing threshold (DFT)-based neurons**, which process input spikes in two stages:
> 1. **Membrane potential accumulation**:$$V_j^l(t) = \sum w_{ij}^l \cdot \mathrm{ReL}(I_i^{l-1}(t))$$
> 2. **Dynamic Spike firing (DFT mechanism)**:The firing time $ t_j^l $ is determined by direct/standard firing or no firing based on a dynamic threshold $\theta^l[t]$
> Note that the spatial ERF is similar to the original formulation. The temporal ERF captures how the timing of input spikes influences the output spike timing.
> As the spike time-to-first-arrival (TFA) is a primary parameter to describe the output spikes, we rewrite the temporal ERF in the manuscript Eq.(7) as:
> $$\mathrm{ERF}_\mathcal{T}(s^{\ell}, \tau) =\sum\frac{\partial \mathcal{L}}{\partial s^{\ell}[T]} \cdot \frac{\partial s^{\ell}[T]}{\partial (T-\tau)^{\ell}}$$
> $$\frac{\partial s^{\ell}[t]}{\partial t^{\ell}}=\frac{\partial s^l[t]}{\partial \Theta_j^l[t]} \cdot \frac{\partial \Theta_j^l[t]}{\partial V_j^{l}[t]} \cdot \frac{\partial V_j^{l}[t]}{\partial t^{l}} - \frac{\partial s^l[t]}{\partial \Theta_j^l[t]} \cdot \frac{\partial \Theta_j^l[t]}{\partial \theta^l[t]} \cdot \frac{\partial \theta^l[t]}{\partial t^{l}}$$
>
> By setting the gradient stimuli $\frac{\partial \mathcal{L}}{\partial s_{(i,j)}^{\ell}[T]}$ to 1, we can compute the temporal ERF using Pytorch's autograd functionality. This allows us to analyze how the timing of input spikes influences the output spike timing, thus providing insights into how information is encoded in the timing of spikes.

---

> > ### Comment · Reviewer_Yc3k · 2025-08-05
> >
> > The rebuttal adequately addresses my concerns, particularly the equation clarifications and extension to other neuron models. The additional experiments on DVS datasets strengthen the work's generalizability.
> >
> > The paper makes a solid theoretical contribution with ST-ERF and demonstrates practical improvements.

---

> > > ### Author Response · Authors · 2025-08-05
> > > **Response to Reviewer Yc3k**
> > >
> > > Thank you for your thoughtful feedback. We are very pleased that our rebuttal successfully addressed your concerns, particularly appreciating your endorsement of our proposed theoretical framework. We will ensure that these discussions are integrated into the revised manuscript to further improve its quality.

---

### Official Review · Reviewer_cNmF · 2025-07-03

**Clarity:** 4
**Significance:** 3
**Originality:** 3
**Rating:** 5
**Confidence:** 4

**Summary:**

This paper proposes a novel framework called the Spatio-Temporal Effective Receptive Field (ST-ERF) to study the learning behavior of both spatial and temporal domains. It discovers two key patterns: spatial patterns are Gaussian-like, and their influence fades predictably over time. Using these findings, the authors analyze existing spiking transformers and then build two new efficient building blocks (MLPixer and SR blocks). Finally, they test these new blocks successfully on tasks requiring processing long sequences of visual data.

**Questions:**

1. How would ST-ERF behave under other neuron types like Izhikevich or P-LIF? Would the power-law decay pattern still hold?

2. Please provide a clearer explanation of what each subplot (a)-(d) in Figure 5(b) represents. What cause the jagged temporal ERF?

**Ethical Concerns:**

["NO or VERY MINOR ethics concerns only"]

**Final Justification:**

I think the rebuttal did solve my concerns. I would like to see this paper be accepted.

**Limitations:**

Yes.

**Paper Formatting Concerns:**

No.

**Quality:**

3

**Strengths And Weaknesses:**

Strength:
1. ​​Introduces a new way (ST-ERF) to understand both spatial and temporal learning in Spiking Neural Networks, filling an important research gap.
2. ​​Rigorously demonstrates ST-ERF's core findings (Gaussian space, power-law time decay) through comprehensive experiments on synthetic models and real-world tasks.
3. ​​Designs simple but impactful new blocks (MLPixer, SR Block) that significantly improve SNN performance on demanding long-sequence vision tasks.

Weakness:
1. Some of the operator notations in equations (especially Eq. (12), (13)) are overloaded or ambiguously defined, making it harder to parse the exact pipeline.
2. other see questions.

---

> ### Author Rebuttal · Authors · 2025-07-30
>
> > Q1: Some of the operator notations in equations (especially Eq. (12), (13)) are overloaded or ambiguously defined.
>
> A1: We appreciate your feedback regarding the operator notations in equations, particularly Eq. (12) and (13). Here we provide a table summarizing the definitions of the operators used in these equations:
> | Operator | Definition |
> |----------|------------|
> | $\mathrm{Mixer}_{ch}(\cdot)$ | Channel mixer block|
> | $\mathrm{SSC}(\cdot)$ | Spike-driven Seperable Convolution block |
> | $\mathrm{BN}(\cdot)$ | Batch Normalization |
> | $\mathrm{MLP}(\cdot)$ | Multi-layer Perceptron Layer |
> | $\mathbb{SN}(\cdot)$ | Spiking neuron layer |
>
> We have revised the manuscript to clarify the definitions of the operators used in these equations. The Eq.(12) and (13) is supposed to be:
> $$\mathrm{Mixer}_{ch}(\cdot) = \mathrm{BN}(\mathrm{MLP}(\mathbb{SN}(\mathrm{BN}(\mathrm{MLP}(\mathbb{SN}(\cdot))))))$$
>
> $$\mathrm{Mixer}_{ch}(\cdot) = \mathrm{BN}(\mathrm{Conv}(\mathbb{SN}(\mathrm{BN}(\mathrm{MLP}(\mathbb{SN}(\cdot))))))$$
>
>
> We have ensured that all operators are clearly defined and consistently used throughout the manuscript. We hope this clarification addresses your concerns. If you have any further questions or suggestions, please feel free to let us know.
>
> > Q2: ST-ERF behaviors in other spiking neuron models.
>
> A2.1: **Izhikevich's spiking neuron model:**
>
> We conducted the Temporal ERF framework on Izhikevich's spiking neuron model, which is a well-known model in computational neuroscience. The Izhikevich model is defined by the following equations:
>
> $$
> \frac{dV}{dt} = 0.04V^2 + 5V + 140 - U + I,
> \frac{dU}{dt} = a(bV - U)$$
>
> $$if  \space  V \geq v_{threshold}, then \space V \leftarrow c, U \leftarrow U + d$$
>
> where $V$ is the membrane potential, $U$ is the recovery variable, and $I$ is the input current. The parameters $a$, $b$, $c$, and $d$ are constants that define the specific behavior of the neuron. We discretized the equations using Euler's method and implemented the Izhikevich model in PyTorch and Spikingjelly, which can be formulized as follows:
>
> $v(t + \Delta t) \approx v(t) + \Delta t \left[ 0.04v(t)^2 + 5v(t) + 140 - u(t) + I(t) \right]$,
>
> $u(t + \Delta t) \approx u(t) + \Delta t \left[ a(bv(t) - u(t)) \right]$
>
> $V \leftarrow c, U \leftarrow U + d$
>
> Description of each parameter is provided in the following table:
> |Parameter|Description|
> |---------|-----------|
> |$a$|Time scale of the recovery variable|
> |$b$|Sensitivity of the recovery variable to the membrane potential|
> |$c$|Reset value of the membrane potential after a spike|
> |$d$|Increment of the recovery variable after a spike|
> |$\Delta t$ or $tau_{inv}$|Time step for discretization|
>
> By properly setting the parameters, we can acquire the desired spiking behavior as we discovered in the manuscript. The parameter $\Delta t$ primarily controls the time resolution of the simulation, and influences the temporal ERF evolution (same as the parameter $\beta$ in the manuscript). The detailed parameter settings and results are summarized in the following table:
>
> | a | b | c | d | tau_inv | v_threshold | Temporal ERF |
> |---|---|---|---|---------|-------------|--------------|
> | 0.02 | 0.2 | -65.0 | 6.0 | 0.006 | 2.0 | Exponential Decay|
>
> Note that the Izhikevich model can be extremely flexible, and the parameters can be adjusted to achieve various spiking behaviors(e.g. bursting, regular spiking, etc.). And we believe that the exploration of these behaviors using the Temporal ERF framework can provide valuable insights into the dynamics of spiking neural networks.
>
> A2.2: **Parametric LIF:**
>
> We also conducted the Temporal ERF framework on the Parametric LIF model, which is a variant of the LIF model that incorporates learnable parameters to control the dynamics of the neuron. Because it is a variant of the LIF model, the Temporal ERF is computed in a similar way as described in the manuscript, which follows the exponential attenuation of the membrane potential over time.
>
> **Both the results of Izhikevich's spiking neuron model and the Parametric LIF model demonstrate that the Temporal ERF framework can effectively capture the temporal dynamics of spiking neurons, regardless of the specific neuron model used.** The exponential decay behavior observed in both models aligns with our theoretical framework, confirming the robustness and versatility of the ST-ERF approach. We have included these visual results in the updated manuscript to provide a more comprehensive evaluation of the ST-ERF framework across different spiking neuron models.
>
> > Q3: Provide a clearer explanation of what each subplot (a)-(d) in Figure 5(b) represents. What cause the jagged temporal ERF?
>
> A3.1: We apologize for the lack of clarity in the legend of Figure 5(b). In Figure 5(b), we aimed to illustrate the temporal ERF patterns of Spiking CNNs with different spike activations. We employ a 20-layer deep SCNN network to derive temporal ERFs across four distinct surrogate functions. From subfigures (a) to (d), the surrogate functions are as follows: (a): $\mathrm{arctan(\cdot)}$, (b): $\mathrm{Sigmoid}(\cdot)$, (c): $\mathrm{Rect(\cdot)}$, (d): $\mathrm{Poly(\cdot)}$.
>
> In each subfigure, we visualize the temporal ERF patterns at different decay factors. Considering that the spike activations are achieved by LIF neurons, we can describe the forward process as follows:
>
> $\mathbf{v}^{\ell}[t]=\mathbf{h}^{\ell}[t-1]+f({\mathbf{w}^{\ell}},\mathbf{x}^{\ell-1}[t-1]), \text{(Charging function)}$
>
> $\mathbf{s}^{\ell}[t]=\mathbf{\Theta}(\mathbf{v}^{\ell}[t]-\vartheta), \text{(Firing function)}$
>
> $\mathbf{h}^{\ell}[t]=\beta\mathbf{v}^{\ell}[t]- \vartheta \mathbf{s}^{\ell}[t], \text{soft reset}$
>
> Accordingly, the temporal ERF is defined as:
> $$
> \underset{\mathcal{T}}{\mathrm{ERF}}(s^{\ell}, \tau) = \sum_{i,j} \sum_{\hat{i},\hat{j}} \frac{\partial s_{(\hat{i},\hat{j})}^{\ell}[T]}{\partial s_{(i,j)}^{\ell-1}[T-\tau]} = \sum_{i,j} \frac{\partial \mathcal{L}}{\partial s_{(i,j)}^{\ell-1}[T-\tau]}
> $$
>
> where $\mathcal{L}$ is the loss function, $\mathbf{v}^{\ell}[t]$ is the membrane potential, $\mathbf{s}^{\ell}[t]$ is the spike output at layer $\ell$, and $\mathbf{h}^{\ell}[t]$ is the hidden state at layer $\ell$ at time $t$. The decay factor $\beta$ controls the leakiness of the neuron, and $\vartheta$ is the threshold for spiking.
>
> Proven by Appendix D of the original manuscript, the temporal ERF is the temporal effective receptive field decays exponentially with time delay $\tau$, and the decay rate is primarily determined by the membrane potential decay constant $\beta$, i.e., $\underset{\mathcal{T}}{\mathrm{ERF}}(\cdot,\tau) \propto \beta^{\tau}$. The four experiments used decay factors of 2.0, 1.33, 1.2, and 1.14, respectively, which may not be the commonly-said decay factors in many essays. Some amount of works define the LIF model as
>
> $\mathbf{h}^{\ell}[t]=(1-\frac{1}{\tau})\mathbf{v}^{\ell}[t]- \vartheta \mathbf{s}^{\ell}[t], \text{soft reset}$
>
> where $\tau$ is the decay factor. However, in our theoretical framework, we use $\beta=1-\frac{1}{\tau}$ for convenience while ensuring consistent monotonicity between $\beta$ and $\tau$
>
> The shaded regions represent the $\mathrm{mean \pm std}$ deviation across 20 independent experimental trials. The solid lines represent the mean value points of the original data  connected, while the dashed lines represent curves fitted using $y=a \beta^x + b$. Throughout the experiments, we observed that the fitted curves closely followed the trends of the original data, validating our theoretical framework.
>
> A3.2: In the analysis of Temporal ERF, it is essential to ensure that the network avoids vanishing or exploding gradients during backpropagation. Therefore, we meticulously controlled the surrogate gradient functions of neurons, the network’s parameter settings, and the value ranges of input tensors in our implementation to ensure stable capture of Temporal ERF characteristics during backpropagation. Detailed parameter configurations can be found in Appendix E. At the same time, we used PyTorch's automatic differentiation to compute gradients. Due to the inherent limitations of floating-point arithmetic (IEEE 754 standard), numerical errors may occur during computation, leading to jagged fluctuations in the Temporal ERF curve. This phenomenon is common in deep learning, especially when processing high-dimensional data or complex models.

---

> > ### Comment · Reviewer_cNmF · 2025-08-04
> >
> > I think the rebuttal did solve my concerns. I would like to see this paper be accepted.

---

> > > ### Author Response · Authors · 2025-08-04
> > > **Response to Reviewer cNmF**
> > >
> > > We sincerely appreciate your recognition of our work. We will thoroughly revise the manuscript in accordance with your suggestions and those of the other reviewers to enhance its quality.

---

### Note · Authors · 2025-08-13

Here, we would like to reiterate the advantages of the ST-ERF framework proposed in our paper and summarize key reviewer questions to inform the AC’s decision.

This paper presents the Spatio-Temporal Effective Receptive Field (ST-ERF) framework, a novel tool for analyzing Spiking Neural Networks (SNNs) that fills a key research gap and establishes a solid theoretical foundation.
## Core Contributions:
**1. Pioneering Framework**: First ERF extension to SNNs, revealing Gaussian spatial and power-law temporal receptive fields, with broad generality confirmed on Izhikevich and LIF models.

**2. Architecture Diagnosis**: Uses ST-ERF to reveal global receptive field limitations in Transformer-based SNNs (S-ViTs), supported by visual analysis.

**3. New Modules & Gains**: Proposes lightweight MLPixer and SR Blocks to enhance global receptive fields, significantly boosting performance on COCO, ADE20K, DVS classification, and event tracking, integrating advanced ANN ideas into SNN design.
## Reviewer Concerns:
**Authors addressed all reviewer concerns with rigor, substantially improving the paper’s quality and impact.**

**1. Clarity of Formulas and Notations  `[cNmF-1]`.**

We have thoroughly revised the manuscript to clarify the definitions of the operators used in these equations.

**2. Figure Annotation and Explanation Issues: `[cNmF-2]` `[Yc3k-1]` `[Yc3k-2]` `[oAHP-1Q]`.**

We provided persuasive explanations and additional evidence to clarify the meanings of the figures.

**3. Model Applicability, Generalization, Comparisons and Experimental Design: `[Yc3k-3]` `[cNmF-1Q]` `[oAHP-1,2,2Q]` `[X2Er-1,2,3,4]`.**

We explored broader task validation, including DVS and event-based tracking benchmarks, showing the methods are fully generalizable. We conducted ST-ERF analyses on additional neuron types, including Izhikevich, P-LIF, and delayed coding neurons, to demonstrate its applicability to various SNN models.

**4. Biological Relevance and Theoretical Connections: `[Yc3k-1Q,2Q]`.**

We have established explicit links between ST-ERF and major biological theories, including neural oscillation theory, synaptic plasticity (STDP), and compatibility with pulse timing encoding.

**5. Design Motivation and Practical Application: `[X2Er-4]` `[oAHP-3Q]`.**

We clarified the path from ERF analysis to architectural choices and highlighted practical implications for the reviewers.

All reviewer concerns had been addressed, and the four reviewers provided positive feedback.

---

### Decision · Program_Chairs · 2025-09-17

**Decision:**

Accept (poster)

**Comment:**

This paper received four positive reviews, with all four reviewers converging to accept recommendations --- 2 Accept, 2 Borderline Accept.

There was general appreciation for the importance of the problem considered here, the novelty of the solution and the analysis, the strength of the results and the overall presentation.

While there were some concerns raised in the original reviews, they were adequately addressed during the author-reviewer discussion phase. As a result, an accept decision was reached.